



# How Well do Models Represent the Development of Extra-Tropical Cyclones? Evaluation of Two General Circulation Models Against NAWDEX IOP 6 Observations

David L. A. Flack[1*], Gwendal Rivière[1], Ionela Musat[1], Romain Roehrig[2], Sandrine Bony[1],
Julien Delanoë[3], Quitterie Cazenave[3], and Jacques Pelon[3]

[1]Laboratoire de Météorologie Dynamique/IPSL, Ecole Normale Supérieure, PSL Research University, Sorbonne University, École Polytechnique, IP Paris, CNRS, Paris, France
[2]CNRM, Université de Toulouse, Météo-France, CNRS, Toulouse, France
[3]LATMOS-IPSL, CNRS/INSU, University of Versailles, Guyancourt, France
**Correspondence:** Gwendal Rivière (griviere@lmd.ens.fr)

**Abstract.** The dynamical and microphysical properties of a well-observed cyclone from the North Atlantic Waveguide and Downstream impact Experiment (NAWDEX), called the Stalactite cyclone and corresponding to Intensive Observation Period 6, is examined using two atmospheric global circulation models: CNRM-CM6-1 and IPSL-CM6A. The hindcasts are performed in "weather forecast mode", run at CMIP 6 resolution (LR) and c. 0.5° (HR) and initialized during the initiation stage of the

cyclone. Cyclogenesis results from the merging of two relative vorticity maxima at low levels: one is associated with a Diabatic Rossby Vortex (DRV) propagating from the subtropics and the other is initiated by baroclinic development and interaction with a pre-existing upper-level PV cut-off. All hindcasts produce (to some extent) a DRV. However, the second vorticity maximum is almost absent in LR hindcasts because of an underestimated upper-level PV cut-off. The evolution of the cyclone is examined via the quasi-geostrophic $\omega$ equation, which separates the diabatic heating component from the dynamical one at each given

time. In contrast with some previous studies, there is no change in the relative importance of diabatic heating with increased resolution. The analysis also shows that IPSL-CM6A produces a more active cyclone compared to the CNRM-CM6-1 due to stronger diabatism. To examine this further, hindcasts initialized during the mature stage of the cyclone are compared with airborne remote-sensing measurements. There is generally an underestimation of the ice water content in the model compared to the one retrieved from radar-lidar measurements, even when the liquid water content is added. Consistent with the increased

diabatism in IPSL-CM6A compared to CNRM-CM6-1, the sum of liquid and ice water contents is higher in IPSL-CM6A than CNRM-CM6-1 and, in that sense, IPSL-CM6A is closer to the observations. However, ISPL-CM6A strongly overestimates the fraction of super-cooled liquid compared to the observations by a factor of approximately 50.

---

[*]Current Affiliation: Met Office, Exeter, UK



## 1 Introduction

Extra-tropical cyclones are one of the leading hazards in the midlatitudes but their projected behaviour under climate change remains rather uncertain (e.g. Harvey et al., 2012). This uncertainty lies in the location of the extra-tropical cyclones and the intensity and position of the storm track (e.g. McDonald, 2011; Zappa et al., 2013b) rather than in the total number of extra-tropical cyclones (e.g. Finnis et al., 2007; Bengtsson et al., 2009; Catto et al., 2011; Zappa et al., 2013b).

Uncertainties in climate simulations can arise from three different factors: model physics, internal variability, and forcings
(e.g. Hawkins and Sutton, 2009). Therefore, to determine confidence in future projections the historical model climate is compared to observations or re-analyses (e.g. Seiler and Zwiers, 2016). Typically, the representation of cyclones in climate models is considered through statistics, e.g. number, frequency, storm-track position, and cyclone intensity (e.g. Zappa et al., 2013a; Seiler and Zwiers, 2016). These studies generally indicate systematic limitations of coarse-resolution models rarely producing explosively-deepening cyclones, producing too many weak cyclones, and a storm track that is both too zonal and
too far south.

Recently, studies have started to investigate the 3D structure in cyclones (e.g. Catto et al., 2010) and the roles of diabatic heating in climate models (e.g. Willison et al., 2013; Trzeciak et al., 2016; Sinclair et al., 2020). Willison et al. (2013) and Trzeciak et al. (2016) showed that increased resolution, compared to that of the Coupled Model Intercomparison Project (CMIP) models at the time (CMIP 5), was required to improve the representation of the diabatic heating, and hence representation of
the cyclone. This improved representation of diabatic heating could be important as Sinclair et al. (2020) indicated that diabatic processes could become more important in a warming climate. However, the fundamental processes linked to extra-tropical cyclone formation and development need further investigation in global circulation models (GCMs). The fundamental processes linked to factors such as cyclogenesis and cyclone development are hard to examine in full length, free-running, climate simulations and could explain a lack of consideration of this to date. Therefore, to examine the representation of the physical
processes in cyclone formation and development different techniques are required. These techniques include running climate model configurations in "weather forecast mode" (e.g. Phillips et al., 2004), or running short ensemble forecasts (e.g. Wan et al., 2014).

The idea of running climate model configurations in "weather forecast mode" has been around for some time (e.g. Phillips et al., 2004). This idea culminated in the formation of the Transpose - Atmospheric Model Intercomparison Project (T-AMIP)
experiments (Williams et al., 2013). The T-AMIP experiments are primarily used to assess whether any long-term model biases (particularly in the "fast-physics" variables) occur within the first few days of the simulations. It was hoped that, if these biases formed early in the climate simulations, model improvements to reduce those biases could be tested with less computational expense (e.g. Williams et al., 2013; Ma et al., 2013). It was further thought this application could help disentangle the origin of the model biases in a more causal way (e.g. Brient et al., 2019). The T-AMIP experiments have considered factors such as
cloud cover behind fronts in extra-tropical cyclones (e.g. Williams et al., 2013); radiative feedbacks (e.g. Williams et al., 2013; Fermepin and Bony, 2014); 2 m temperature (e.g. Fermepin and Bony, 2014; Ma et al., 2014); precipitation (e.g. Ma et al., 2013; Fermepin and Bony, 2014; Pearson et al., 2015; Li et al., 2018); stratocumulus (e.g. Brient et al., 2019); and for use





alongside random-parameter ensembles to determine structural *vs.* parameter sensitivities in climate simulations (e.g. Sexton et al., 2019; Karmalkar et al., 2019).

The T-AMIP type experiments can also be used as a powerful tool for considering the way climate models represent dynamical processes within weather events. For example, Trzeciak et al. (2016) showed that climate models of resolution T127 (c. 1.1–1.5° resolution at midlatitudes) can represent deep extra-tropical cyclones and their tracks well. This resolution was increased (approximately double) compared to the CMIP 5 climate models and was attributed to an increased importance of the diabatic heating. Like Trzeciak et al. (2016), we consider the dynamical representation of extra-tropical cyclones, and the im-

pact of resolution, in climate models. However, we focus on a single, well-observed, cyclone during the Intensive Observation Period (IOP) 6 of the North Atlantic Waveguide and Downstream impact EXperiment (NAWDEX) field campaign (Schäfler et al., 2018), which is called the "Stalactite" cyclone. This cyclone is initiated from the interaction of two features that occur predominantly on scales smaller than the gridlength of current climate models. The main deepening phase is characterized by interaction of the surface cyclone with successive synoptic-scale upper-level troughs. Here, we answer the following questions

on the representation of the cyclone in climate models to provide further insights into whether climate models are producing cyclones for the correct reasons:

1. How well do climate models represent the two stages of the Stalactite Cyclone?

2. What are the relative roles of diabatic and dynamic processes in the development of the Stalactite Cyclone?

The NAWDEX field campaign occurred in September–October 2016 with the aim of making targeted observations of pro-

cesses that numerical atmospheric models poorly represent (Schäfler et al., 2018). These observations would then be used to help determine how well the models can represent these processes (e.g. Maddison et al., 2019; Oertel et al., 2019). The observations taken during the field campaign allow for an extra question to be asked in this study:

3. Can microphysical observations made during the field campaign give any useful information about the climate model's performance?

To our knowledge this study is the first time that a climate model is compared with flight data taken during a field campaign, without nudging of analyses into the simulation, and it is only feasible because of the T-AMIP protocol.

The questions asked here are of particular interest for the Stalactite cyclone as it influences the development of a blocking anticyclone over Scandinavia and marks the transition between a North Atlantic Oscillation (NAO) positive regime and a Scandinavian blocking regime over the North Atlantic European sector. Therefore, it is a particularly useful case to determine

the capabilities of our current climate models.

The remainder of this paper has the following layout. The key features of the Stalactite Cyclone are discussed in Section 2. The GCMs, experimental setup, observations, and diagnostics are described in Section 3. The Stalactite cyclone's representation in the two GCMs are discussed within Section 4. A summary is made in Section 5.



## 2  NAWDEX IOP 6: The Stalactite Cyclone

The Stalactite Cyclone corresponds to IOP 6 of the NAWDEX field campagin (Schäfler et al., 2018). It was an explosively-deepening cyclone that initially formed at 1800 UTC on 29 September 2016 (Fig. 1a) off the coast of Newfoundland (c. 56° W, 45° N; Fig. 1b). Cyclogenesis occurred as a result of the merging of two vorticity maxima at low levels (Fig.1c): The northern maximum over Newfoundland is formed via baroclinic interaction with the upper-level PV cut-off. The PV cut-off extended down to the surface like a stalactite (hence the name of the cyclone). The southern maximum corresponds to a Diabatic Rossby

Vortex (DRV). A DRV corresponds to an isolated positive PV anomaly rapidly travelling eastward in a moist and baroclinic region. To determine if this diabatic precursor is a DRV we use the criteria set by Boettcher and Wernli (2013). The authors define 6 criteria to detect DRV: (i) SLP minimum detection, (ii) a minimum PV value at 850 hPa (0.8 PVU averaged near the SLP minimum), (iii) substantial low-level baroclinicity, (iv) fast propagation (250 km in 6 hours), (v) sufficient moisture and (vi) very weak upper-level forcing. All these criteria are met in ECMWF analysis which confirms the identification of a DRV.

It was formed on 27–28 September off the coast of Florida and South Carolina (not shown). Our analysis showed that it was probably produced from a mesoscale convective system, as confirmed by satellite images showing cold brightness temperature (see e.g. Fig. 1e).

The two low-level precursors merge into a single cyclonic vorticity maximum in a vortex roll-up by the subsequent analysis (not shown). The initial cyclogenesis phase led to a short deepening stage over 18 h as the cyclone travelled east past New-

foundland. The cyclone underwent a second, more substantial, deepening as a result of an interaction with a large-scale region of high PV at upper levels as the cyclone began to cross the North Atlantic. This region is marked by multiple regions of high PV ("B" and "C" in Fig. 1d) that are successively injected in the upper-level disturbance ("A") and interacts with the Stalactite cyclone. The deepening occurred at a rate of 24.1 hPa in 24 h and so, for this latitude, meets the criterion set in Sanders and Gyakum (1980)[2] to be classified as an explosively-developing cyclone. The explosive deepening occurred between 1800 UTC

September – 1800 UTC 1 October (Fig. 1a) where the structure developed and showed an increasing cloud head and well defined fronts (Fig. 1f). During the interaction with the second large-scale trough cyclonic wave breaking occurred and the cyclone re-curved towards Greenland (Fig. 1b). On reaching the coast of Greenland cyclolysis (i.e. cyclone decay) occurred; the cyclone had filled in by 0000 UTC 4 October. The cyclone did not make landfall and so had no societal impacts. However, the cyclone posed an interesting challenge for the operational numerical weather prediction models as the cyclone occurred

during a regime transition from an NAO positive regime to a Scandinavian blocking regime which dominated the North Atlantic European sector for the rest of the field campaign (e.g. Schäfler et al., 2018; Maddison et al., 2019). Correspondingly, there was a reduction in the forecast skill (Schäfler et al., 2018).

To determine whether the climate models are correctly simulating the Stalactite Cyclone three criteria are developed from its lifecycle.

---

[2]A deepening rate of 1 hPa h$^{-1}$ for 24 h multiplied by $\sin(\phi)/\sin(60)$ to adjust to the appropriate latitude to make it equivalent to at least 1 bergeron, where $\phi$ the latitude





1. Initial cyclogenesis occurs as a result of the merger of a DRV and another near-surface cyclonic vortex located further north and associated with baroclinic interaction with an upper-level PV cut-off.

2. A main deepening phase associated with large-scale troughs is present.

3. A minimum pressure deepening rate of 24 hPa in 24 h during the secondary deepening phase.

If all of these criteria are met then the climate models are able to correctly represent the Stalactite cyclone. The climate models and experimental setup used are discussed in the following section.

## 3  Models, Observations, and Diagnostics

In this section we discuss the model setup and experimental protocol of the T-AMIP experiments (Section 3.1), the observations used (Section 3.2), and the diagnostics considered (Section 3.3). We also compare our simulations against the European Centre for Medium Range Weather Forecasting (ECMWF) analysis.

### 3.1  Models and Experimental Setup

We use the atmospheric components of two climate models run in "weather forecast mode" to represent T-AMIP-style experiments: the CNRM-Cerfacs (Centre National de Recherches Météorologiques - Centre Européen de Recherche et de Formation Avancée en Calcul Scientifique; Voldoire et al., 2019) and the IPSL (Institut Pierre Simon Laplace; Boucher et al., 2020) climate models. Both climate models recently contributed to CMIP 6 (Eyring et al., 2016), and here we make use of the same model versions and configuraions.

The CNRMCM61 atmospheric component is based on the version 6.3 of the global atmospheric model ARPEGEClimat (Roehrig et al., 2020). It is a spectral model derived from the ARPEGE/IFS (Integrated Forecast System) numerical weather prediction model developed jointly by MétéoFrance and the ECMWF. It is run with 91 vertical levels, and at two different horizontal resolutions. The first of these resolutions corresponds to a T127 truncature (c. 150km globally; hereafter denoted as ARPEGE-LR) and the second to a T359 truncature (c. 50km; hereafter ARPEGE-HR). The CNRM-CM6-1 convection scheme is based on the work of Piriou et al. (2007) and Guérémy (2011). The longwave radiation scheme corresponds to the global circulation model (GCM) version of Rapid Radiation Transfer Model (RRTM; Mlawer et al., 1997), whilst the ECMWF/IFS cycle 32 radiation scheme is used of the shortwave component (Fouquart and Bonnel, 1980; Morcrette et al., 2008). The microphysics scheme follows the work of Lopez (2002), and the turbulence scheme that of Cuxart et al. (2000). The model output is then converted onto a 1.4 ° and 0.5 ° latitude/longitude grid for ARPEGE-LR and -HR respectively for the purposes of analysis.

The IPSL-CM6A atmospheric component, known as LMDZ6A (Hourdin et al., 2020), is run with 79 vertical levels, and also at two different horizontal resolutions. The first of these resolutions is the CMIP 6 resolution (2.5°×1.2°: IPSL-CM6A-LR and hereafter denoted as LMDZ-LR) and a higher resolution (IPSL-CM6A-HR and hereafter denoted as LMDZ-HR). The IPSL-CM6A-HR configuration utilises the zoom function of LMDZ, in which the resolution over part of the domain is increased





compared to the rest in a variable resolution configuration. For our simulations the zoomed domain is centred at (40° E, 55° N) with a resolution equivalent to 0.33° and the resolution decreases away from the centre. This results in a resolution of approximately 0.5° over the North Atlantic and approximately 1.1° in other parts of the domain. The convection scheme is based on Rochetin et al. (2014), the shortwave radiation is an extension of Fouquart and Bonnel (1980) to six bands, and the

longwave radiation scheme is the GCM version of RRTM (Mlawer et al., 1997). The microphysics scheme follows Madeleine et al. (2020) with a new representation of low clouds (Hourdin et al., 2019) and the surface scheme follows Cheruy et al. (2020).

For both models hindcasts are initiated at 0000 UTC on 27–29 September, and 1–2 October 2016 from the ECMWF analysis. For ARPEGE microphysics state variables and turbulent kinetic energy were initialised to zero and aerosols are prescribed

from a present-day climatology. On the other hand, in LMDZ model state variables not defined in the analysis are set to zero, alongside the aerosols. Sea surface and sea ice cover are also from the ECMWF analysis. All hindcasts are performed out to a leadtime of T+10 d. For both of the models, and all hindcasts, output data is interpolated onto a pressure grid in the vertical, every 25 hPa, from 1000 hPa to 100 hPa. Furthermore, output is also produced using the CFMIP (Cloud Feedback Model Intercomparison Project) Observation Simulator Package (COSP; Bodas-Salcedo et al., 2011) for radar reflectivities

from Cloudsat to be compared with the observed aircraft-borne radar reflectivities from the NAWDEX field campaign.

We restrict the number of hindcasts to take into account the impact of the overall synoptic situation at the time being largely unpredictable (e.g. Schäfler et al., 2018). This is confirmed by the hindcasts initiated on 27 and 28 September at low resolution not producing the Stalactite Cyclone (not shown).

As with all T-AMIP experiments our simulations are initialised with non-native initial conditions as they are initialised

with ECMWF analysis. However, initialising from non-native initial conditions can lead to an adjustment period before the model returns to its natural trajectory and balance. This adjustment period is referred to as the "initial shock" (e.g. Klocke and Rodwell, 2014). The initial shock can be seen through the presence of gravity waves throughout the domain or by comparing with simulations started from the native analysis. Therefore, to ensure the same diagnosis of initial shock in both models the presence (or lack) of numerical artefacts (such as small-scale disturbances and gravity waves) and their persistence was

examined. In considering the first 12 h of each simulation no numerical gravity waves or small-scale disturbances were noted in any of the model runs (not shown), thus initial shock does not appear to be significant. As a precautionary measure, we do not analyse the hindcasts prior to T+18 h.

To consider the uncertainty associated with our results we examine the hindcats at different leadtimes. We do not create an ensemble of different initial conditions (i.e. initial conditions from different forecasting centres) as this would produce similar

results given the validity period of T-AMIP protocol (e.g. Fig. 1 in Klocke and Rodwell, 2014). Furthermore, experiments initialised from the Météo-France analyses yield similar results to those initialised from the ECWMF analyses (not shown). Given that ARPEGE and LMDZ have different dynamical cores and parametrizations, some amount of uncertainty in the results can be obtained by comparing the models. The results presented within this paper are consistent as long as a cyclone resembling the Stalactite cyclone is produced in the hindcasts, regardless of leadtime. Therefore, for consistency, all the plots

produced when discussing the life cycle of the cyclone are from the 00 UTC 29 September initialisation, and observation





comparisons (Section 4.4) are considered from the hindcast initialised at 00 UTC 1 October 2016, unless otherwise stated. The first time (00 UTC 29 September) has been chosen because it is just prior to cyclogenesis and the precursors already exist in the analysis for all resolutions. Hence, it is a well-suited initial time to study the whole life cycle of the cyclone. The second time (00 UTC 1 October) has been chosen, to make comparisons with the observations, because the cyclone is required to be

in (or as close as possible to) the observed location to allow the different features of the cyclone to be comparable.

## 3.2  Observations

During the NAWDEX field campaign, four aircraft equipped with remote sensing and in-situ instruments were operated and among them the French SAFIRE Falcon aircraft from 1 October to 15 October (Schäfler et al., 2018). The SAFIRE Falcon made two flights to observe the Stalactite cyclone on 2 October 2016: F6 (towards Greenland) and F7 (south of Iceland; Fig. 1b). The

second flight (F7) was directly into the cyclone in the ascending branch of the associated warm conveyor belt, as opposed to the first flight (F6) which considered the warm conveyor belt outflow, so in the main manuscript we focus on F7. A comparison with F6 data can be found in the supplementary material and produces similar results to F7 comparisons. The first leg of F7 (the most eastern one) was chosen because there was an overpass with CloudSat-CALIPSO track at 14:09 UTC which allows us to assess observation uncertainties by comparing airborne and spaceborne measurements. The payload onboard the SAFIRE

Falcon included a 95-GHz Doppler cloud radar and a high-spectral resolution Doppler lidar capable of measuring at 355, 532 and 1064 nm (e.g. Delanoë et al., 2013). Measurements by these two instruments allow the retrieval of ice water content thanks to the variational algorithm of Delanoë and Hogan (2008) updated by Cazenave et al. (2019). The combination of radar and lidar further allows for the identification of the phase of the particles to be identified (e.g. super-cooled liquid, ice, liquid, etc.) using principles outlined in Delanoë and Hogan (2010). Furthermore, Doppler-derived windspeeds, and radar reflectivities are

also used from these instruments. The retrievals come from both radar products only (RASTA) and a combined radar and lidar product (RALI) to allow for uncertainty in the measurements to be taken into account.

    To ensure a fair comparison between the observations and the model, the observations are first converted onto the model grid. This conversion assumes a linear relationship between the speed of the aircraft and distance travelled by the aircraft. On the other hand, the model output is converted onto the flight path using the nearest gridpoint to the flight path to create a

"virtual" flight path. Other interpolation methods were tested but had limited impacts on the conclusions (not shown).

    The modelled ice water content (IWC) is compared throughout the entire "virtual" flight track, and also by applying the observation mask (to help determine the impact of positional errors). Furthermore, the model IWC is defined in two ways: "potential" IWC (equivalent to cloud ice plus snow) and "maximum" IWC [equivalent to cloud ice plus snow plus liquid water content (LWC)]. The inclusion of LWC in the "maximum" IWC is to take into account the impact of mixed-phase clouds and

super-cooled liquid. On the other hand, the radar reflectivites are compared without applying the mask due to the output from the COSP simulator occurring in contour frequency altitude diagram form rather than exact radar reflectivity values. Since COSP radar simulator has been developed to be compared to ClouSat radar reflectivity, a comparison between the CloudSat and airborne radar reflectivities is also inserted in the supplementary material to assess observation uncertainties.





The comparisons are made at the most appropriate time with respect to the model cyclone rather than the observation time.
This framework takes into account any delay in the cyclone formation in the climate models (Sec. 4), and ensures that the modelled cyclone has a structure that is as similar as possible to the analysis. In reality, for the hindcast that is compared against the observations in this study (initiated at 00 UTC 1 October 2016) neither timing nor positioning adjustments are required.

### 3.3 Vertical Motion and Baroclinic Conversion Budgets

Extra-tropical cyclone evolution can be considered through many methods. For example, the surface pressure tendency equation (e.g. Fink et al., 2012), through potential vorticity framework (e.g. Davis et al., 1993), or the quasi-geostrophic (QG) vertical motion ($\omega$) equation (e.g. Sinclair et al., 2020). Here, as in Sinclair et al. (2020), we consider the evolution of cyclones through the QG $\omega$-equation. We also take this a step further and consider the energetics of the system through the baroclinic conversion ($BC$). All our results using these methods are considered in a cyclone-relative framework.

The QG $\omega$-equation, that includes diabatic heating and the $\beta$ term, can be written in terms of the so-called $\mathbf{Q}$ vector following Hoskins et al. (1978) and Hoskins and Pedder (1980). We use the formulation of Holton (2004) that includes the diabatic heating too:

$$\left(\sigma\nabla^2 + f_0^2\frac{\partial^2}{\partial p^2}\right)\omega_{QG} = -2\left(\nabla\cdot\mathbf{Q}\right) + f_0\beta\frac{\partial v_g}{\partial p} - \frac{R}{c_p p}\nabla^2 J, \tag{1}$$

for

$$\mathbf{Q} = -\frac{R}{p}\begin{pmatrix} \frac{\partial \mathbf{u_g}}{\partial x}\cdot\nabla T \\ \frac{\partial \mathbf{u_g}}{\partial y}\cdot\nabla T \end{pmatrix},$$

where $\sigma$ is the static stability (which is obtained by temporally averaging the temperature across the lifetime of the Stalactite cyclone), $f_0$ is a reference coriolis parameter, $\beta$ is the beta term in the coriolis forcing, $p$ is the pressure, $R$ is the specific gas constant, $c_p$ is the specific heat, $J$ is the rate of heating per unit mass, $\mathbf{u_g}$ is the geostrophic wind vector, $T$ is the temperature, $x$ and $y$ are the positions in the meridional and zonal direction, respectively, and $\omega_{QG}$ is the vertical velocity obtained from the
QG $\omega$-equation (1). In the rest of the paper, the geostrophic wind is used to compute the $\mathbf{Q}$ vector, but we have checked that using the full horizontal wind components do not change the results so much.

Equation (1) allows us to distinguish between the dynamical and diabatic contributions to the vertical motion in the cyclone. Physically the $\mathbf{Q}$ vector and the $\beta$ terms represent the dynamical components of the flow and the Laplacian of the rate of heating per unit mass represents the diabatic heating. A friction term can also be added to Eq. (1). However, as the friction term
is (at least) an order of magnitude smaller than all of the other terms (not shown) it is neglected.

The third term on the right-hand side of Eq. (1) can be split into components from all of the different model physics parametrizations (e.g. convection, radiation, and large-scale heating from condensation) as follows

$$J = J_c + J_r + J_{lscp}...,$$





where the subscripts represent the initials of the parametrization they represent, e.g. $c$ is convection, $r$ is radiation, and $lscp$ is

large-scale heating due to condensation/large-scale cooling due to evaporation (i.e. latent heating).

     To solve Eq. (1) the 3D Laplacian is inverted using Liebmann successive over-relaxation with boundary conditions such that $\omega$ is zero at 1000 hPa, 100 hPa, and all horizontal boundaries. The vertical motion is computed for every 25 hPa in the vertical. The inversion is not fully accurate due to the QG assumption. However, on average most of the modelled vertical motion is recovered using this method. Furthermore, the timing of the development of the inverted $\omega$ ($\omega_{QG}$) matches that of the model

$\omega$. Further discussion of the differences between $\omega_{QG}$ and the model $\omega$ are discussed in Sec. 4.3. The $\omega_{QG}$ can be split into dynamic ($\omega_{dyn}$) and diabatic ($\omega_{diab}$) components as follows

$$\left(\sigma\nabla^2 + f_0^2\frac{\partial^2}{\partial p^2}\right)\omega_{dyn} = -2\left(\nabla\cdot\mathbf{Q}\right) + f_0\beta\frac{\partial v_g}{\partial p}, \tag{2}$$

and

$$\left(\sigma\nabla^2 + f_0^2\frac{\partial^2}{\partial p^2}\right)\omega_{diab} = -\frac{R}{c_p p}\nabla^2 J. \tag{3}$$

Inversion of the two previous equations allows to separate the contribution of dynamical and diabatic processes in the vertical motion. Such a decomposition provides further insights into the development of the cyclone by determining the balance between these processes in the evolution of the cyclone. Vertical velocity intervenes in different key terms of the classical equations for the development of extratropical cyclones such as in the stretching term of the relative vorticity equation or in the baroclinic conversion term of the kinetic energy equation. In the present study, we adopt the energetic framework and compute

the baroclinic conversion from eddy potential energy to eddy kinetic energy within the extra-tropical cyclone (e.g. Orlanski and Katzfey, 1991; Rivière and Joly, 2006). The baroclinic conversion is proportional to the vertical heat flux and can be written as

$$BC = -h\omega'\theta', \tag{4}$$

where $h = (R/p)(p/p_s)^{R/C_p}$ where $p_s$ is the surface pressure and $\theta$ is the potential temperature. Primes denote the difference

from the 5-day temporal average of that quantity centered over the lifecycle of the Stalactite cyclone. Some tests have shown that the results are rather insensitive to the definition of the temporal average as long as it is made over an interval equal or longer than the life cycle of the cyclone to suppress the signal associated with the cyclone.

     The baroclinic conversion term is mainly positive in areas following the cyclone trajectory (Rivière and Joly, 2006; Rivière et al., 2015). It can be approximated by replacing the vertical velocity by its quasi-geostropic formulation Eq. (1). As $\omega_{QG}$

is split up into its dynamical and diabatic components ($\omega_{dyn}, \omega_{diab}$) following Eqs. (2) and (3), the approximated baroclinic conversion using the quasi-geostrophic equation can be written as:

$$BC_{QG} = -h\omega_{QG}\theta' = -h\omega_{dyn}\theta' - h\omega_{diab}\theta'. \tag{5}$$

     It is worth noting that $\theta'$ remains identical in all of the components in Eq. (5). Therefore, the decomposition of the baroclinic conversion into dynamical and diabatic terms only results from that of the vertical velocity. Also, the temporal means of $\omega$





and $\omega_{QG}$ being small, the primes can be suppressed on those variables. This equation has been kept in this form for simplicity and so will naturally lead to some contamination of the diabatic and dynamic parts within the final results due to $\theta'$ remaining constant.

## 4   Representation of the Stalactite Cyclone

Throughout this section the dynamical and diabatic representation of the Stalactite cyclone is discussed. The usual metrics of
minimum pressure evolution and cyclone track are considered in Section 4.1. An in-depth consideration of the cyclogenesis and development occur in Sections 4.2 and 4.3, respectively. The two climate models are compared to the flight observations and discussed in relation to diabatic heating in Section 4.4.

### 4.1   Pressure Evolution and Track

The representation of the Stalactite Cyclone is first considered via an overview of the cyclone through its track and minimum
sea level pressure evolution (Fig. 2). The LR hindcasts both produce a rapidly deepening cyclone: 24 hPa in 24 h in ARPEGE-LR and 38 hPa in 24 h in LMDZ-LR. However, this deepening is delayed by 24 h compared to the analysis, and the initial cyclogenesis is not as intense as in the analysis. This weaker cyclogenesis results in an initially weaker cyclone compared to the analysis in both models (Fig. 2a). However, the explosive deepening in LMDZ-LR compensates for the lack of initial deepening and results in a cyclone with the same intensity as the observations. On the other hand, ARPEGE-LR has the same
secondary deepening strength as the analysis so produces a weaker cyclone.

The cyclone track also differs from the analysis. The difference occurs 18 h into the hindcasts. The two LR hindcasts produce a track that is too far south and has a later re-curvature so the cyclone track occurs further east compared to the analysis (Fig. 2b). The eastward shift in the track agrees with the global weather forecasts prior to 29 September 2016 (e.g. Maddison et al., 2020). Given the rapid divergence of the forecast track from the analysis, differences in the cyclogenesis could
be one aspect leading to the track occurring too far east. The cyclogenesis being important for the cyclone track is partially corroborated by the track representation having improved (i.e. no eastward shift), regardless of resolution, after the cyclone appears in the initial conditions (not shown).

As expected, increasing the resolution improves the representation of the track, timing and intensity of the Stalactite Cyclone. The LMDZ-HR hindcast, like the LMDZ-LR hindcast, produces a deepening rate of 38 hPa in 24 h. The initial deepening is
well represented in LMDZ-HR whereas it is absent in LMDZ-LR. However, the latter run has a very rapid deepening during the second stage of development which compensates the absence of the first stage. This compensation has resulted in a cyclone with a similar intensity to the analysis but with a delay of roughly 1 day in the deepening. The ARPEGE-HR and ARPEGE-LR hindcasts are also marked by rapid deepening phases with deepening rates being close to 24 hPa in 24 h. However, the cyclones are less deepened in ARPEGE runs compared to LMDZ runs (compare the blue and red curves). There is a significant delay in
the deepening phase of the cyclone in ARPEGE-LR compared to ARPEGE-HR, similar to the difference between LMDZ-LR and LMDZ-HR, which is also due to the quasi-absence of the initiation stage in the lower-resolution run. Thus both models are





able to produce an explosively-deepening cyclone at the two resolutions considered. The main differences to the representation of the Stalactite cyclone compared to the analysis, on initial inspection, appear to be within the cyclogenesis phase of the cyclone and the different deepening rate of LMDZ compared to ARPEGE. These two aspects are examined further within the following subsections.

## 4.2 Cyclogenesis

The cyclogenesis of the Stalactite cyclone occurs on the mesoscale as the merging of two low-level vorticity precursors: a DRV coming from the subtropics and a vortex located further north baroclinically interacting with an upper-level PV cut-off (Fig. 1c). In the present section, we analyze the representation of the two precursors and their subsequent merging in the different simulations. The same vorticity fields as in Fig. 1c are shown in Fig. 3 for the different simulations. Figures 4 and 5 show the baroclinic conversion at T+18 h for both ARPEGE and LMDZ respectively and help identify the mechanisms behind the two precursors for the Stalactite Cyclone.

### 4.2.1 The Diabatic Rossby Vortex

Criteria of DRV introduced by Boettcher and Wernli (2013) and recalled in section 2 have been analyzed in the different simulations. The two HR hindcasts fit all the criteria of a DRV, which shows that 50 km grid spacing is enough to represent the DRV. Criteria (i), (iii), (v) and (vi) (see section 2) are also met in LR runs but the LR hindcasts do not meet the criteria (ii) and (iv) based on PV intensity (for both) and propagation speed (LMDZ-LR). However, although the criteria of Boettcher and Wernli (2013) are not met in the LR configurations it is encouraging to see that they produce a qualitative representation of a DRV, that is approaching the criteria, despite the coarse resolution of the models and the mesoscale nature of this self-sustaining phenomenon. The identification of the southern precursor as being a DRV is confirmed by the baroclinic conversion of Figs. 4 and 5 which show that the diabatic component is almost equal to the total in the vicinity of the vortex and that the dynamical component is negligible. The DRV is more active in LMDZ compared to ARPEGE as the associated heating rate reaches higher values in LMDZ compared to ARPEGE (e.g., compare Figs. 4c and Figs. 5c). Vertical cross sections of the heating rates across the DRV indicate that its structure extends throughout the atmospheric column (Fig. S3 in the supplementary material) confirming the impression left by the satellite image (Fig. 1e). The cross sections also confirms the stronger diabatic heating in LMDZ compared to ARPEGE in the vicinity of the DRV.

### 4.2.2 Formation of the northern precursor via baroclinic interaction with the PV cut-off

More important differences appear between LR and HR runs in the representation of the northern precursor. First, in the LR hindcasts, the vorticity of the northern precursor is much smaller than the vorticity of the DRV precursor (2.4 × smaller in ARPEGE-LR and 3.3 × in LMDZ-LR) whereas it is only slightly smaller in HR runs (ratio of 1.6 in ARPEGE-HR and 1.3 in LMDZ-HR) similarly to the analysis. Second, the two LR runs (Figs. 3a,c) have a more zonal PV cut-off than in the analysis (Fig. 1c) and in the two HR runs (Figs. 3a,c). Third, the low-level northern vorticity maximum moves to the east of the cut-off




**Table 1.** Distance and PV criteria from Boettcher and Wernli (2013) for identifying DRVs between T+18 and T+24 from the hindcast initialised on 00 UTC 29 September. The PV criterion is based on a minimum PV value when averaged at the minimum MSLP location and the eight surrounding grid boxes, which is set to 0.8 PVU. The distance criterion is based on a minimum distance travelled by the vortex in 6 hours, which is 250 km. When the threshold is reached a ✓ is present, otherwise a ×. The DRV is described as quantitative if all the thresholds are reached and qualitative if the thresholds are not reached

. Only criteria where at least one of the hindcasts do not meet the criteria are shown, all other DRV criteria from Boettcher and Wernli (2013) are met by all hindcasts.

| Model | PV (T+18) [PVU] | PV (T+24) [PVU] | 850 hPa PV criterion [✓ / ×] | distance [km] | distance criterion [✓ / ×] | DRV type [Qualitative/Quantitative] |
|---|---|---|---|---|---|---|
| ECMWF (1.1°) | 0.94 | 1.18 | ✓ | 370.6 | ✓ | Quantitative |
| ARPEGE-LR | 0.65 | 0.72 | × | 302.4 | ✓ | Qualitative |
| ARPEGE-HR | 1.45 | 1.74 | ✓ | 328.3 | ✓ | Quantitative |
| LMDZ-LR | 0.51 | 0.68 | × | 138.6 | × | Qualitative |
| LMDZ-HR | 2.77 | 2.54 | ✓ | 264.5 | ✓ | Quantitative |

in the HR runs and analysis, which is typical of strong baroclinic interaction, whereas it stays to the south of the cut-off in LR runs (Figs.3a and c).

In contrast with the DRV, the northern precursor is a mixture of diabatic and dynamic processes as shown by the baroclinic conversion rates of Figs. 4 and 5. The vertical cross sections of Fig. 6 show that the dynamical component is mainly centred at upper levels, but with an equivalent-barotropic structure. This suggests that the northern precursor is forced by the vertical velocity associated with the PV cut-off which is characteristic of type-B cyclogenesis (Petterssen and Smebye, 1971). In LR hindcasts, the dynamical forcing has a smaller vertical extent and is more spread out than the HR hindcasts. Also, the dynamical

forcing in LR hindcasts is located further east than the diabatic forcing (Figs. 6b and c) while the two forcings add more to each other in HR hindcasts (Figs.6e and f; see also figure S2 for ARPEGE). Both of the two forcings increase with resolution by a factor of more than five in the two models. However, the peak values of the diabatic baroclinic conversion exhibit larger increase than those of the dynamical baroclinic conversion with resolution during the formation of the northern precursor (Figs. 5b,c,e,f and 6b,c,e,f) but as seen in next section, it is more the reverse that occurs during the rest of the life cycle. To

conclude, the northern precursor is rather badly represented in LR compared to HR hindcasts because the less intense, and more spatially diluted, PV inside the cut-off induces a weaker, and more spread out, dynamical forcing. There is also a weaker spatial correlation between the dynamical and diabatic vertical forcing in the LR compared to HR hindcasts. An additional factor is the more active diabatic forcing in HR hindcasts in the vicinity of the northern precursor. So both the dynamical and diabatic terms, and their overlap, improve with resolution and it is difficult to determine which component matters most.





### 4.2.3 Merging of the two precursors

For the hindcasts shown here the merger of these two different precursors differs in timing from the analysis and between resolutions. The higher resolution configurations (although delayed by 6 h compared to the ECMWF analysis) merge the DRV and upper-level dynamical precursor 12–18 h earlier than the LR runs (not shown). For LMDZ-LR, there is even no merging of the two precursors. The delay or absence of interaction between the two precursors has an impact on the track of the cyclone which was systematically located too far east in the LR runs (Fig. 2b). There are two factors to explain the delayed or missed merging. One is the more rapidly eastward propagation of the DRV in HR than LR runs (see e.g., the 2° more eastward shift of the DRV in HR compared to LR runs in Fig. 3), which is consistent with a stronger latent heating in the former runs. The second is the low-level northern precursor and the upper-level cut-off are moving less rapidly eastward in HR runs (not shown). This can be partly explained by the difference in longitude of the dynamical forcing between LR and HR hindcasts (compare Figs. 6b and e). The more rapid propagation of the DRV and less rapid motion of the northern precursor explain why the DRV is more able to catch up the northern precursor in HR runs as in the analysis.

To conclude on cyclogenesis, the LR hindcasts struggle to correctly represent the initiation of the cyclone because they miss the initial deepening of the northern small-scale low-level vortex and the roll-up of the merging two low-level vortices around the PV cut-off. However, the unexpected result is that the LR hindcasts are able to reproduce the behavior of the DRV rather well, albeit with a smaller propagation speed.

### 4.3 Main Deepening

The main focus of this section is the main deepening stage of the Stalactite Cyclone. Like the cyclogenesis phase the main deepening phase shall be considered by analysing the baroclinic conversion and averaging it over a $10° \times 10°$ area centred on the maximum baroclinic conversion closest to the minimum pressure of the Stalactite Cyclone. (Fig. 7) or by computing its local maximum (Fig. S3). The averaged quasi-geostrophic baroclinic conversion is roughly reduced by two thirds in magnitude compared to that directly calculated from the model $\omega$ but is consistent across the models for the timing of the growth and decay of the cyclones. As previously discussed there is a delay in the maximum deepening in LR runs compared to HR runs, about 1 day in LMDZ and half a day in ARPEGE.

In the cyclone average values shown in Fig. 7, the two stages of cyclone development are well separated: (i) the initial cyclogenesis stage occurring on 29 September (Section 4.2), and (ii) the main development stage that is dominated by the presence of a large-scale trough and an explosively-developing cyclone. The initiation stage is clearly dominated by diabatic processes. On the other hand, during the main deepening stage, the dynamical processes begin to be more important, and more so in the HR hindcasts compared to the LR hindcasts. In the HR runs the dynamical term is even larger than the diabatic term during the whole main deepening stage. It is also clear in the LR hindcasts that there is a delay in the dynamical processes compared to the diabatic processes, suggesting a delayed forcing by the large-scale upper-level trough. Therefore, there is an increased importance of the dynamic term relative to the diabatic term with increased resolution. This ratio consistency is true for both the maximum (Fig. S3) and average values (Fig. 7) in both models and leadtime. This ratio consistency disagrees with





the previous studies of Willison et al. (2013) and Trzeciak et al. (2016). Whilst we use a different diagnostic from both of these previous studies, so the result could be subject to method sensitivities, case studies or development stage. Indeed, in the previous section we did note a more important role played by the diabatic term during the northern precursor formation when the resolution is increased but here we show the reverse occurs during the main deepening stage of the cyclone.

Considering the dynamical processes in more detail (Figs. 8 and S4) helps to indicate the reason for the delay in the maximum deepening in the LR hindcasts compared to the analysis and the HR hindcasts. The figures clearly show that the HR hindcasts are closer to the analysis than the LR hindcasts. On 1 October 00 UTC, an upper-level PV signature is clearly visible above the surface cyclone in HR hindcasts while it is not the case for the LR hindcast for which the cyclone is still mainly a DRV. However, the PV injection coming from the large-scale region of high PV located to the northeast into the upper-level disturbance interacting with the surface cyclone is delayed. In the analysis, some PV injection has already occurred (white areas of 5 to 7 PVU in Fig. 8a) but is just starting in the HR runs (the blueish areas of 3 to 5 PVU in Fig. 8c). The situation in ARPEGE-HR on 1 October 12 UTC (Fig. 8f) resembles more that of the ECMWF approximately 6 hours earlier (not shown), with the cyclonic wave breaking being more advanced in the ECMWF analysis (Fig. 8d) compared to ARPEGE-HR (Fig. 8f). Several studies have shown that the PV of the upper-level trough baroclinically interacting with a surface extratropical cyclone tends to advect the cyclone polewards (Rivière et al., 2012; Oruba et al., 2013; Coronel et al., 2015). Therefore the sooner nonlinear interaction of the cyclone with the large-scale upper-level PV reservoir and the sooner roll-up of the two features around each other explains the sooner deviation of the cyclone track to the north and the more westward position of the track in the analysis than in the hindcasts. For the HR hindcasts, the delay is a maximum of 6 hours and the eastward shift is minimal while for LR hindcasts the delay is about 24 hours and the eastward shift is much more marked.

### 4.4    Interpretation of the difference between the models and comparison with Aircraft Observations

As previously said, to have cyclone features roughly at the same place in the models as in the observations, simulations initiated at 00 UTC 1 October 2016 are analyzed in the present section. In other words, the idea is to have the dynamical features roughly at the same place in the observations and simulations to be comparable.

#### 4.4.1    Diabatic heating in the models

Figures 2a and 7 show that LMDZ produces a more active cyclone than ARPEGE. This activity is hypothesised to be linked to the vertical structure of diabatic heating within the Stalactite Cyclone given the similarities between the large-scale troughs in both models. To examine this hypothesis distributions of the $\omega_{QG}$ are considered over the cyclone (Fig. 9). Whilst Fig. 9 shows the hindcasts initiated at 00 UTC 1 October 2016 similar results occur for the hindcasts initiated at 00 UTC 29 September 2016 (not shown). Figure 9 shows that $\omega_{QG}$ increases with increased resolution (c.f. Figs. 9a,c and b,d). Also, the larger ascents mainly arise from diabatic processes (c.f. Figs. 9e-g to i-l), which is particularly obvious for LR hindcasts. Concerning the differences between models, stronger ascents occur in LMDZ than ARPEGE when looking at $\omega_{QG}$ (compare Fig. 9a to c and b to d). In LR hindcasts, the largest difference between ARPEGE-LR and LMDZ-LR occurs near 600-700 hPa and is clearly attributed to the more frequent strong ascents of the diabatic component (Figs. 9e,g) while the dynamical component partly





offsets this difference (Figs. 9i,k). The largest values of $\omega_{QG}$ in LMDZ-HR compared to ARPEGE-HR (Figs. 9b,d) come from both the diabatic and dynamic components with the former component being the dominant one especially for levels above 600 hPa.

To conclude, the difference in the mid-tropospheric vertical motion is mainly a result of diabatic processes ($\omega_{diab}$; Figs. 9f,h)

and indicates that there is more diabatic heating in LMDZ compared to ARPEGE. Thus it is likely that this larger diabatic heating in the mid troposphere accounts for the more active cyclone in LMDZ compared to ARPEGE. The terms that dominate the heating profiles both in ARPEGE and LMDZ are the large-scale condensational heating and convective terms (not shown). Thus it is likely that observations of microphysical properties of the Stalactite Cyclone could be used to qualitatively determine which model has the better heating rates or structure. These comparisons are considered next.

**4.4.2    Microphysical properties in the models and observations**

To determine whether observations of microphysical properties from field campaign flights can provide information on the underlying diabatic heating the Stalactite Cyclone hindcasts are compared with flights F6 and F7 (Fig. 1b) of the SAFIRE Falcon during the NAWDEX field campaign. To ensure an as fair as possible comparison between the flight and hindcasts, hindcasts initiated at 00 UTC 1 October 2016 are compared to the observations.

The large-scale features of the cyclone (such as windspeed) are well represented in both climate models, and at all resolutions, with there only being a small shift (given the rather coarse resolution of the models) in the probability density function toward smaller values by less than $5 \, \mathrm{m \, s^{-1}}$ (not shown). This comparison provides confidence in the large-scale features of the cyclone. Therefore, microphysical features can be further considered. Figure 10 shows bi-variate histograms of the total IWC for F7 from two observation platforms: RASTA (Fig. 10a) and RALI (Fig. 10f). There are larger values of IWC in RASTA

compared to RALI because the lidar (being sensitive to smaller ice particles and smaller quantities of ice) information in RALI leads to a reduction of IWC compared to RASTA. However, both platforms show the same shape with increasing values of IWC to around $600 \, \mathrm{hPa}$ and then a uniform distribution until around $800 \, \mathrm{hPa}$, below which the instruments no longer detect ice clouds. The two retrieved IWC histograms provide an indication of uncertainty of the observations, which is useful to be compared with model outputs.

The model contribution to Fig. 10 consists of four rows, the first two rows showing "potential" IWC (cloud ice + snow) while the last two rows "maximum" IWC (IWC + liquid water content). Comparing the first two rows (Figs. 10b–e and g–j) with the observations shows an underestimation of the model IWC. This underestimation is by a factor of 3–4, similarly to what Rysman et al. (2018) found when comparing observations and WRF simulations of Mediterranean systems. Furthermore, the peak of the model IWC distribution occurs at 700–750 hPa, 100–150 hPa lower than in the observations. There are

small improvements with resolution: the HR simulations have larger IWC content throughout, and particularly aloft and in the maximum values. Furthermore, there are differences between the models. The first difference is that the IWC values of LMDZ-LR are more dispersed than those of ARPEGE-LR suggesting a larger number of ice clouds at this altitude in LMDZ-LR (e.g. see Figs. 10b,d, Figs. 10g,i, and the differences in Figs. 11a,b). The stronger values at upper levels in LMDZ are more inline with the values given by the observations than ARPEGE (c.f. Figs. 10a–j). However, although LMDZ may be better at





representing the IWC content at upper levels, the overall shape of the distribution is better in ARPEGE compared to LMDZ. Indeed, the decreased IWC from 600 hPa to 300 hPa is better represented in ARPEGE. Applying the observation mask to the models (Figs. 10g–j) brings the frequencies more in line with the observations compared to without the mask by removing all the lowest values seen in the no-mask statistics. This is due to the fact that the instruments are not sensitive to very small IWC contents and also the models do not create discontinuity in IWC between cloudy and clear-sky regions. Besides, the comparison

between the mask (Figs. 10g–j) and no-mask (Figs. 10b–e) values implies that there are very small IWC values in the model outside of the observed region (particularly for ARPEGE-LR) indicating the horizontal structure of the cyclone is reasonable.

     Is the underestimated IWC in the models due to the underestimated liquid-to-solid transition for cold temperatures or to the underestimation of condensates as a whole? To answer this question the liquid water content (LWC) below 273 K is added to the IWC to create the last two rows (total IWC; Figs. 10k–r). Adding the LWC makes limited difference to either of the

ARPEGE hindcasts (Figs. 10k,l,o,p) suggesting that either there are fewer LWC points added or the LWC points added have a small magnitude. On the other hand, adding LWC into the LMDZ definition drastically changes the shape and increases the values of total IWC at lower levels (Figs. 10m,n,q,r). The LMDZ distributions have been changed to the extent that the shape now shows much better agreement with the observations than when the LWC was not taken into account. These changes in LMDZ are also apparent within Fig. 11, where the addition of LWC in LMDZ produces consistently larger values of total IWC

at all heights compared with ARPEGE, although this model difference is reduced at increased resolution (Figs. 11c and d). The much larger IWC+LWC in LMDZ compared to ARPEGE over all the levels is consistent with the larger diabatic heating shown in Figs. 9e-h.

     Given the change by the inclusion of LWC in the definition of the IWC it is useful to know the proportion of ice, mixed phase, and super-cooled liquid points that make up these distributions. Here, we arbitrarily define ice points in the model to be

those where the LWC component of the total IWC is less than 1% and "pure" super-cooled liquid to be points where the LWC component is greater than 99% of the total IWC, all other points are mixed phase. These results are compared with those points defined as super-cooled liquid, mixed phase and ice retrieved IWC from RALI measurements. To ensure a fair comparison between ice and super-cooled liquid water the "pure" values are combined with the mixed phase values. Table 2 shows that whilst the combined ice points exceed that of the observations (particularly for ARPEGE) the values are not unreasonable.

However, when the combined super-cooled liquid water is considered the models significantly over-estimate the amount of super-cooled liquid points, with the smallest difference being a factor of 24 and the largest by a factor of 47. Considering Table 2 alongside the earlier discussion of the impact of adding LWC shows that the super-cooled liquid water being added to ARPEGE is of a smaller magnitude than that of LMDZ – a result confirmed by changing the threshold and seeing large decreases in the LWC for ARPEGE (with thresholds up to 10% for the definition of ice) and the values remaining constant in

LMDZ. It is also worth noting that although the LR hindcasts are more largely underestimating the IWC than the HR hindcasts, they are closer to the observations (in shape) than the HR hindcasts in the percentage of super-cooled water.

     Radar reflectivities confirm the strong underestimation of IWC in the hindcasts (Fig. 12). For instance, below 5 km altitude, most of the reflectivity values are in the range 0 to 15 dBz in the observations, -10 to 5 dBz in ARPEGE, and -15 to 0 dBz in LMDZ. The smaller values reached by LMDZ compared to ARPEGE is probably due to the larger percentage of





**Table 2.** The fraction of points within F7 that have values deemed as super-cooled liquid, mixed phase and ice. "MAX"=IWC+LWC, combined super-cooled liquid = super-cooled liquid + mixed phase, and combined ice = ice + mixed phase. The hindcasts are initiated at 00 UTC 1 October 2016 and uses the nearest-gridpoint to the flight path from the two times surrounding the flight path (12 and 15 UTC 2 October 2016; T+36–39 h).

|  | Observations [%] | LMDZ-LR [%] | LMDZ-HR [%] | ARPEGE-LR [%] | ARPEGE-HR [%] |
|---|---|---|---|---|---|
| Super-cooled liquid (LWC > 0.99("MAX")) | 1.5 | 1.2 | 0.5 | 0.0 | 0.0 |
| Mixed phase (0.01("MAX") < LWC < 0.99("MAX")) | 0.2 | 72.8 | 79.7 | 41.4 | 64.6 |
| Ice (LWC < 0.01("MAX")) | 98.3 | 26.0 | 19.8 | 58.6 | 38.4 |
| Combined super-cooled liquid | 1.7 | 74.0 | 80.2 | 41.4 | 61.6 |
| Combined ice | 98.5 | 98.8 | 99.5 | 100.0 | 100.0 |

liquid hydrometeors which induce smaller reflectivities than ice. It also confirms that the LR hindcasts outperform the HR hindcasts and ARPEGE is better than LMDZ in terms of shape of the IWC distribution. Despite a systematic underestimation of reflectivity at all levels, the ARPEGE-LR reflectivity exhibits the closest shape to the observations compared to the other three hindcasts, with a majority of values between 0 and 5 dBZ below 5 km and then a rather constant decrease of reflectivity values with altitude above 5 km as in the observations.

Finally, to be confident in the above results, additional figures are presented in the supplementary material. Figures S5 to S7 support the above findings by doing the same analysis along flight F6. Also, a comparison between RALI and CloudSat-CALIPSO measurements has been made along the common path of flight F7 and the A-train. The CloudSat reflectivities have similar structure and similar amplitude as the RALI reflectivities (Fig. S8c,d). The DARDAR and RALI target classifications agree for the large pictures and the main discrepancies originate from the time shift and the higher noise in CALIOP backscatter

and the lower sensitivity of RASTA close to the surface. This explains why the supercooled layers detection is consistent but the mixed phase attribution is slightly different due to the radars sensitivity (Fig.S8e,f). Despite these differences, regions of combined super-cooled liquid (supercooled plus mixed phase) are rather similar which gives confidence in the above conclusions.

To conclude, LMDZ produces more IWC which is associated with a more intense latent heating than ARPEGE. In that

sense, it is closer to the observations. However, the ratio between liquid *vs.* solid species contributing the IWC is less realistic in LMDZ than ARPEGE. Hence, it is worth noting that whilst the IWC can provide some information about the diabatic heating, caution is needed in interpreting the results as it does not provide complete information to be able to determine which of the two models produce the better heating compared to reality. However, the microphysical observations from flights during





field campaigns are still useful in helping to identify the deficiencies of each model and determine what processes are linked

in the models and why one of the models produces a more active cyclone compared to the other.

## 5   Summary

The representation of the Stalactite Cyclone in the two atmospheric components of the CNRM and IPSL climate models (CNRM-CM6-1 and IPSL-CM6A) has been examined in detail. The two models are run at two resolutions: one at a rather coarse resolution of approximately 150–200 km (LR hindcasts) and the other at a higher resolution of approximately 50 km

(HR hindcasts). The T-AMIP protocol is used to determine how well the climate models can represent the physical processes linked to the Stalactite cyclone and how well it compares to flight observations made during the NAWDEX field campaign. The protocol also gives us valuable insight into the formation of the Stalactite Cyclone.

   Figure 13 shows a schematic of the many stages of the Stalactite Cyclone: from initiation as a DRV initiated from a mesoscale convective system (point 0) through the merger of the DRV (point 1) and a dynamical forcing factor (point 2) at cyclogenesis

(point 3), to its rapid deepening associated with strong diabatic forcing throughout the column and interaction with multiple, embedded, upper-level high PV regions (point 4), and comparisons with the observations (point 5) round to cyclolysis. There are differences between each of the models and with the analysis at each of these points and these are summarised in the main results below. The points are numbered based on the schematic (Fig. 13).

1. All hindcasts produce a DRV to some degree of accuracy: LR hindcasts produce a qualitative DRV whereas HR hindcasts
produce a quantitative DRV that meet the criteria of Boettcher and Wernli (2013) (Table 1).

2. All models produce an upper-level potential vorticity cut-off. However, due to its fine-scale structure, the cut-off is not as intense nor as deep in the LR hindcasts as in the HR hindcasts and analysis. Therefore, the LR hindcasts produce a weaker northern precursor.

3. Due to the above the initial deepening associated with the vortex roll-up between the two precursors at cyclogenesis is
weaker in LR hindcasts. However, the initial deepening is better represented when the resolution increases. This reduced initial deepening implies that LR versions cannot fully (dynamically) represent the Stalactite Cyclone.

4 (a). All hindcasts produce an explosively-deepening cyclone during the mature stage, although this is delayed by 24 h in LR hindcasts. Both CNRM-CM6-1 and CNRM-CM6-1-HR hindcasts agree with the analysis and produce a 24 hPa deepening in 24 h. However, both IPSL-CM6A-LR(-HR) hindcasts are more active than CNRM-CM6-1(-HR) and produce a
deepening of 38 hPa in 24 h. For the IPSL-CM6A-LR hindcast this excessive deepening rate compensates for the lack of initial deepening during cyclogenesis.

4 (b). Diabatic heating extends throughout the troposphere during maximum deepening for both models, but is larger in IPSL-CM6A-LR(-HR) compared to CNRM-CM6-1(-HR). Increasing the resolution does not increase the relative contribution of diabatic heating to the main deepening of the cyclone (when averaging across the cyclone) unlike in previous studies



(e.g. Willison et al., 2013; Trzeciak et al., 2016). Instead there are local increases in the diabatic heating which are particularly important for the northern precursor at cyclogenesis (Figs. 6 and S2).

5 (a).    Both models and resolutions underestimate the ice water content from flight observations, even when super-cooled liquid water is taken into account, by a factor of 3–4 in agreement with Rysman et al. (2018) and this even occurs for mesoscale models (e.g. Mazoyer et al. *submitted*). However, the shape of the vertical distribution of ice water content is in good
agreement for CNRM-CM6-1(-HR) when ice water content is taken into account. The IPSL-CM6A-LR(-HR) hindcasts only come into agreement for the shape of the distribution with the observations when super-cooled liquid water is added to the ice. In that case, when all condensates are considered, the IPSL-CM6A-LR(-HR) model presents larger values compared to CNRM-CM6-1(-HR) over the whole troposphere, and in that sense, is closer to the observations. This larger content of condensates is associated with larger diabatic heating, larger vertical velocities, and hence provides
an explanation for the larger deepening rate in IPSL-CM6A-LR(-HR) compared to CNRM-CM6-1(-HR).

5 (b).    Both models appear to substantially over-estimate the amount of super-cooled liquid water content in the cyclone. This comes mainly as a result of an increased number of mixed phase gridpoints. Although the values of super-cooled liquid being added to the model are lower in CNRM-CM6-1(-HR) compared to IPSL-CM6A-LR(-HR).

Thus, returning to the originally proposed questions and criteria for the correct representation of the Stalactite Cyclone
the evidence suggests that climate models, when they are run at a coarse resolution, cannot represent the initial stage of the Stalactite cyclone but they can produce the main deepening during the mature stage. The results also indicate that improvements in dynamical processes are as (if not more) important as improvements in diabatic processes with increasing resolution. The results further show that microphysical properties can be used, with caution, qualitatively to provide indirect information on the diabatic heating in climate models. Therefore the flight observations provide (albeit not complete) an interesting insight
into whether the climate models are producing the correct heating. This last topic is currently being investigated further by the authors with respect to the downstream impact of extra-tropical cyclones in climate models on subsequent ridge building.

Although the present results only apply for this particular case study[3], the results have important implications and show areas that warrant further investigation. Firstly, it shows that the T-AMIP protocol is useful for considering the physical mechanisms that occur within cyclones and their interaction with dynamics. Secondly, it shows that increasing resolution does help with
the representation of cyclones such that within the next few years, when climate models will be regularly run at c. 50 $\mathrm{km}$, many synoptic-scale features of the atmosphere will be dynamically well represented. Finally, and arguably most critically, it warns that although climate models may produce similar cyclones they can be doing so for very different reasons and these reasons are likely to have an influence upon other areas of the climate system and the response of model cyclones to climate change. We recommend that further research occurs into the partition of super-cooled liquid water, mixed phase and ice water
in models (and the influence this has on cyclone representation) and further comparisons with observations are made in all regions as this will have a strong influence on the development of microphysical schemes in climate and weather prediction models. Therefore, whilst signs are encouraging for future versions of climate models, caution is still needed when considering

---

[3] A second cyclone (the following cyclone; IOP 7 of NAWDEX) related to the future work shows the same results (not shown).




current simulations of future climate scenarios and the impact of extra-tropical cyclones, particularly for regional impact-based studies.

*Data availability.* Data is available by contacting either D. Flack at david.flack1@metoffice.gov.uk or the corresponding author.

*Supplement.* The supplement related to this article is available online at: https://doi.org/10.5194/wcd-0-1-2020-supplement.

*Author contributions.* All authors contributed to the writing and editing of the manuscript as well as the scientific discussions. DLAF produced the first draft and conducted the model analysis. IM performed the IPSL-CM6A simulations and RR the CNRM-CM6-1 simulations. GR and SB designed the study. JD, QC and JP provided the observational datasets.

*Competing interests.* There are no competing interests present.

*Acknowledgements.* This work is part of the DIP-NAWDEX (DIabatic Processes in the North Atlantic Waveguide and Downstream impact EXperiment) project which is funded by the Agence Nationale de la Recherche (ANR) under grant number ANR-17-CE01-0010-01. The funding for IM for running the LMDZ T-AMIP experiments was from the Eurpean research Council grant number 694768. The LMDZ simulations were performed using the HPC from GENCI-IDRIS (grant 0292). The airborne measurements and the SAFIRE Falcon flights received
direct funding from IPSL, Météo-France, INSU-LEFE, EUFAR-NEAREX and ESA (EPATAN, contract n° 4000119015/16/NL/CT/gp). The authors wish to acknowledge use of the Ferret program for analysis and Figs. 1–8 and Figs. S1–S4 in this paper. Ferret is a product of NOAA's Pacific Marine Environmental Laboratory. (Information is available at http://ferret.pmel.noaa.gov/Ferret/). We further acknowledge the use of imagery from the NASA Worldview application (https://worldview.earthdata.nasa.gov), part of the NASA Earth Observing System Data and Information System (EOSDIS).





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





**Figure 1.** An overview of the Stalactite Cyclone. a) the ECMWF analysis minimum pressure evolution (black), solid red lines indicate the initiation phase, dashed red lines the maximum deepening phase, dotted red line the point of minimum pressure. The star represents the time of panel c and e and the triangle the time for panels d and f. b) the track of the Stalactite Cyclone, the start and triangle represent the timing of panels c,e and d,f respectively. The magenta dashed line is the flight path of the SAFIRE Falcon-20 flight 6 and the solid is for flight 7. c) the ECMWF analysis of the 250 hPa PV > 2 PVU (contoured) and 850 hPa relative vorticity (shaded) at cyclogenesis (18 UTC 29 September) and d) as for c) but just before maximum deepening at 12 UTC 1 October. The bold lines are to indicate the PV signature of different PV regions interacting with the cyclone. "A" is upper-level signature of the Stalactite Cyclone at that time, "B" is the second PV region to interact with the cyclone, and "C" is the third. The colour scale applies to both c and d. e) A visible satellite image from MODIS on 29 September indicating the Stalactite cyclone at cyclogenesis. The brightness temperature has been overlaid and saturated at 50%. f) the visible satellite imagery from MODIS on 1 October 2016. The satellite images are produced with the courtesy of NOAA Worldview (https://worldview.earthdata.nasa.gov/).



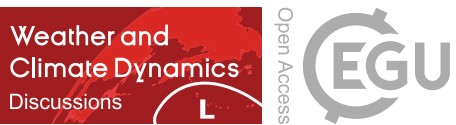

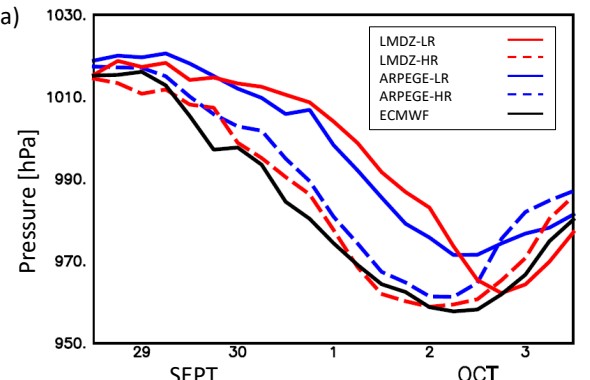

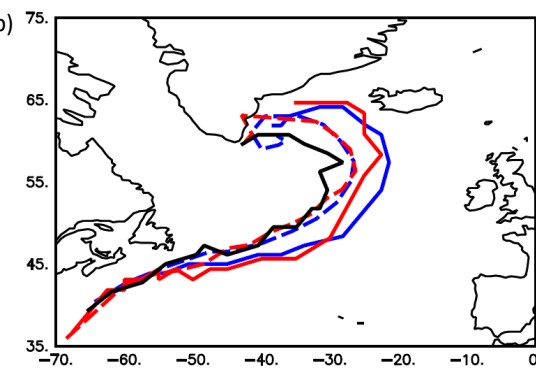

**Figure 2.** An overview of the Stalactite Cyclone: a) the minimum pressure evolution and b) the cyclone track. The ECMWF analyses are in black, LMDZ hindcasts in red, and ARPEGE hindcasts in blue. The LR hindcasts are the solid lines and the HR hindcasts are dashed. All hindcasts are initiated at 00 UTC 29 September 2016.



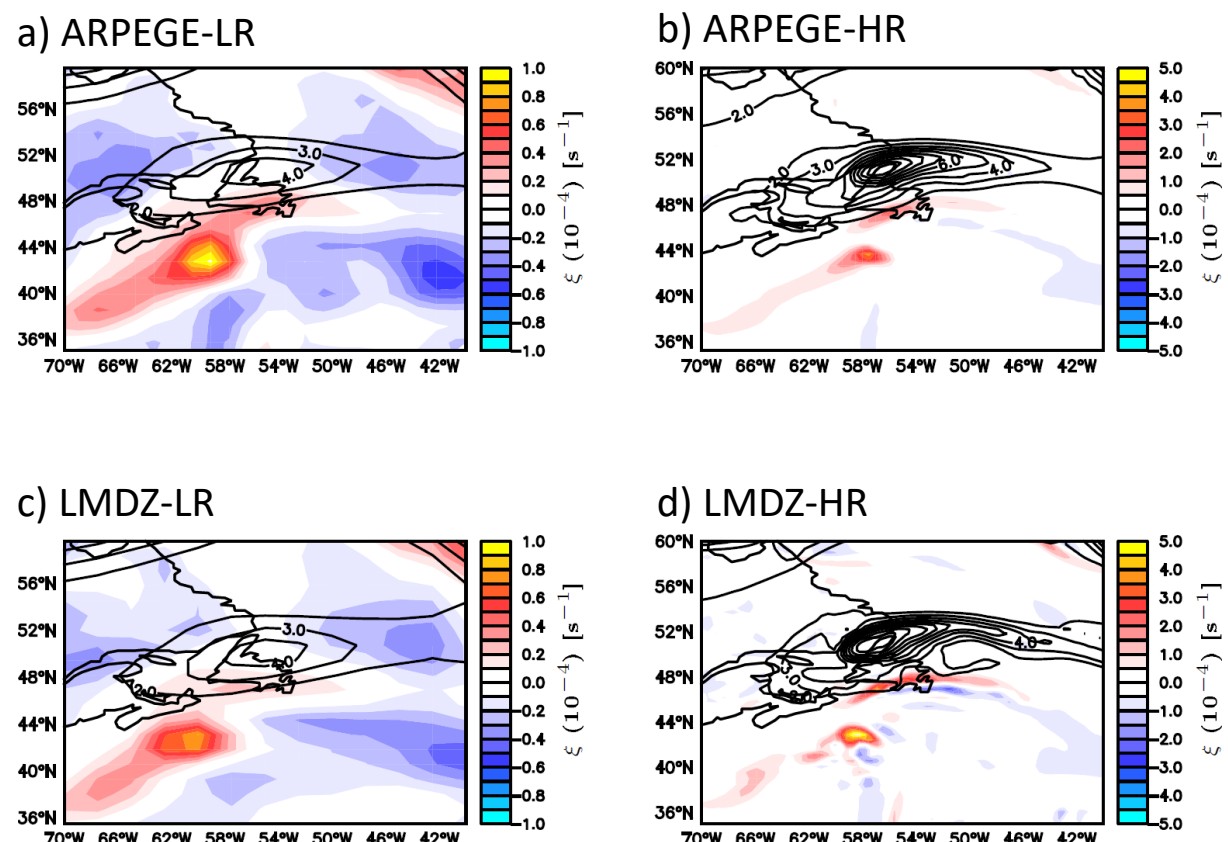

**Figure 3.** Hindcasts for the cyclogenesis of the Stalactite Cyclone at 18 UTC 29 September (T + 18 h) for hindcasts initiated at 00 UTC 29 September. The 250 hPa PV above 2 PVU (contoured every 1 PVU) and the 850 hPa relative vorticity (shaded) for a) ARPEGE-LR, b) ARPEGE-HR, c) LMDZ-LR, and d) LMDZ-HR. The colour scale is different between LR and HR runs.



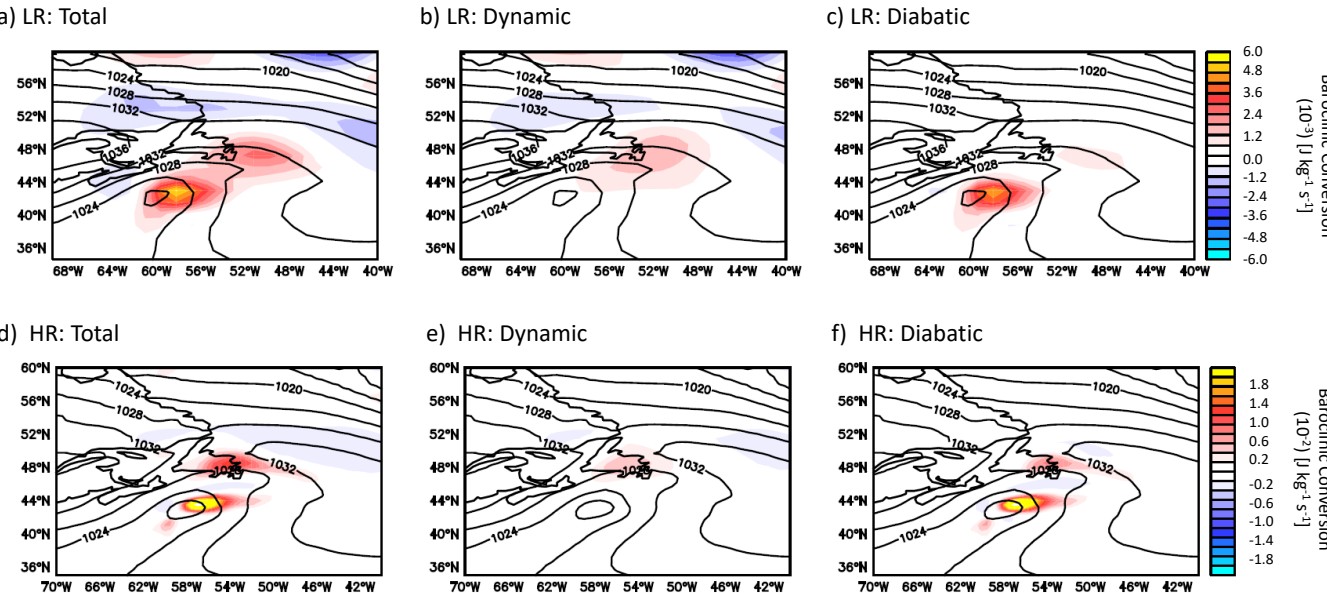

**Figure 4.** Vertically-averaged baroclinic conversion between 850 hPa and 300 hPa (shaded) and mean sea level pressure (contoured) at 18 UTC 29 September 2016 (T + 18 h from hindcast initation at 00 UTC 29 September 2016) for ARPEGE hindcasts. a–c) ARPEGE-LR hindcast and d–f) ARPEGE-HR hindcast. a,d) total (dynamic + diabatic) baroclinic conversion; b,e) baroclinic conversion from dynamical processes; c,f) baroclinic conversion from diabatic processes. The colour scales refer to each row.

.





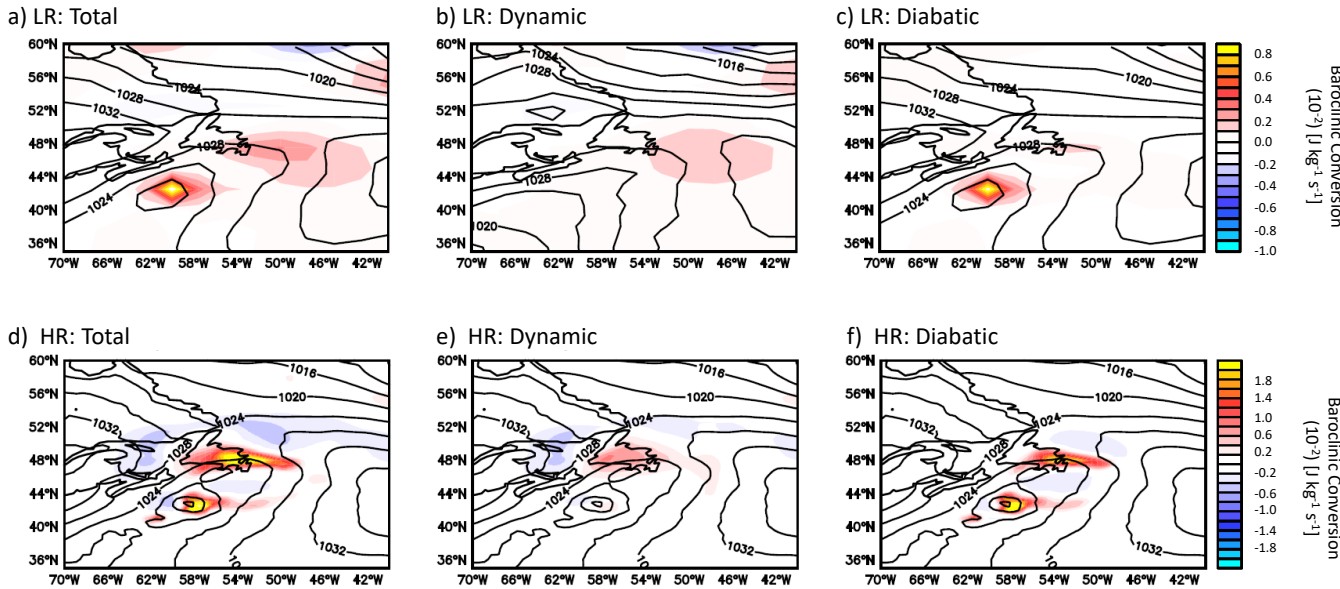

**Figure 5.** As in Fig. 4 but for LMDZ-LR (and -HR) hindcasts.

.



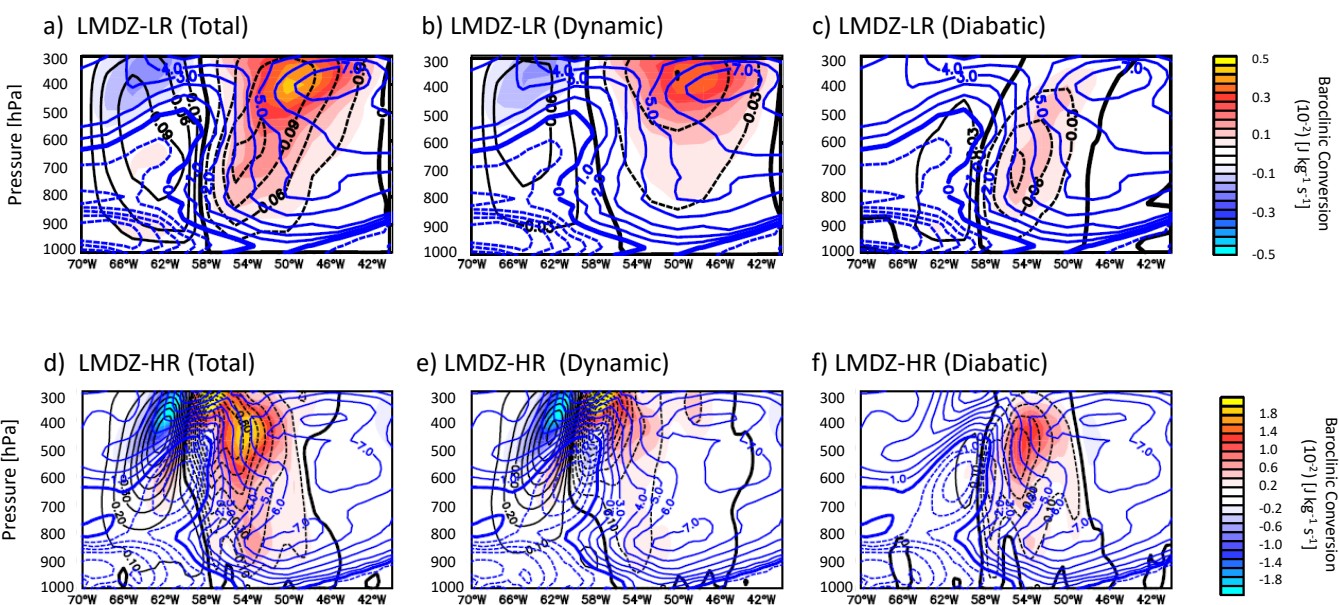

**Figure 6.** A vertical cross-section averaged across the northern precursor in the LMDZ hindcasts at 18 UTC 29 Septmeber 2016 (T+18 h). The baroclinic conversion (shaded), potential temperature anomaly (blue contours) and inverted $\omega$ (black contours) for a,d) total baroclinic conversion; b,e) the baroclinic conversion due to dynamic processes only; and c,f) the baroclinic conversion from diabatic processes. a–c) LMDZ-LR and d–f) LMDZ-HR. Note that the colorscales and contours are different between the LR and HR runs

.



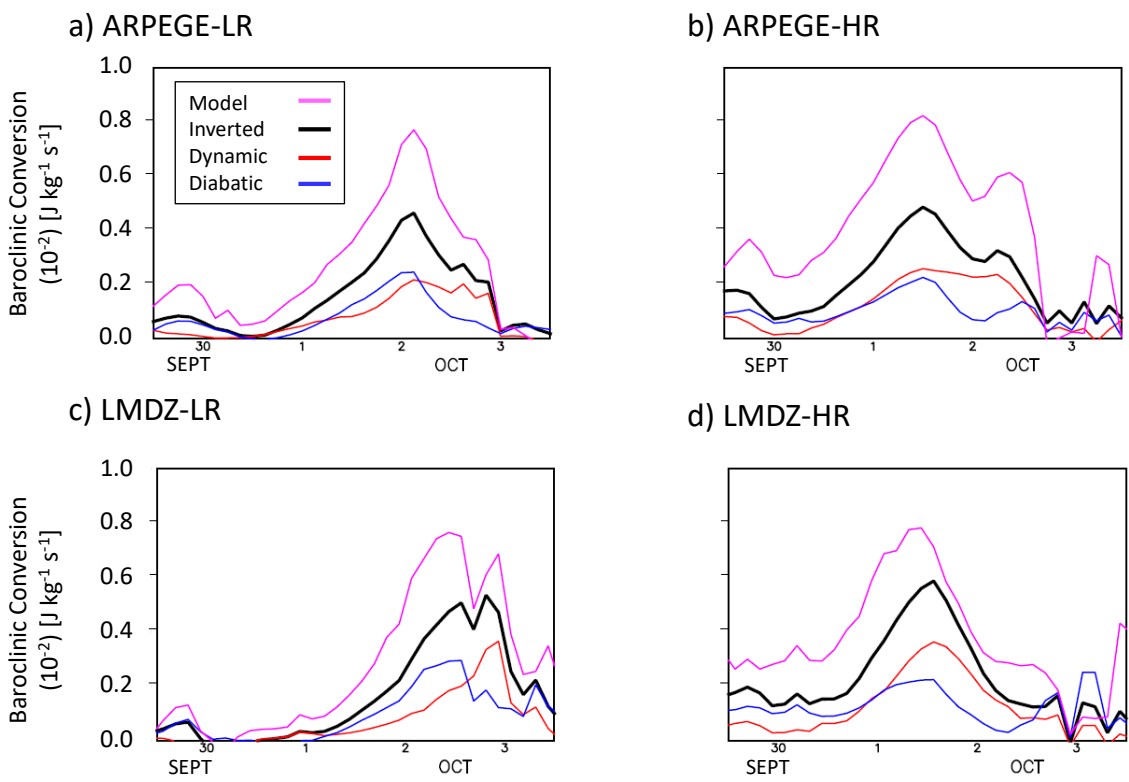

**Figure 7.** The evolution of the average baroclinic conversion in a $10° \times 10°$ box around the maximum baroclinic conversion that is closest to the minimum pressure of the Stalactite cyclone. For a) ARPEGE-LR, b) ARPEGE-HR, c) LMDZ-LR, d) LMDZ-HR. The magenta line is for the baroclinic conversion calculated with the model $\omega$; the black line is the total inverted $\omega$; the red line the inverted $\omega$ from dynamical processes; and the blue line the inverted $\omega$ from diabatic processes. All hindcasts were initiated at 00 UTC 29 September 2016. Maximum point values of baroclinic conversion are shown in Fig. S3.



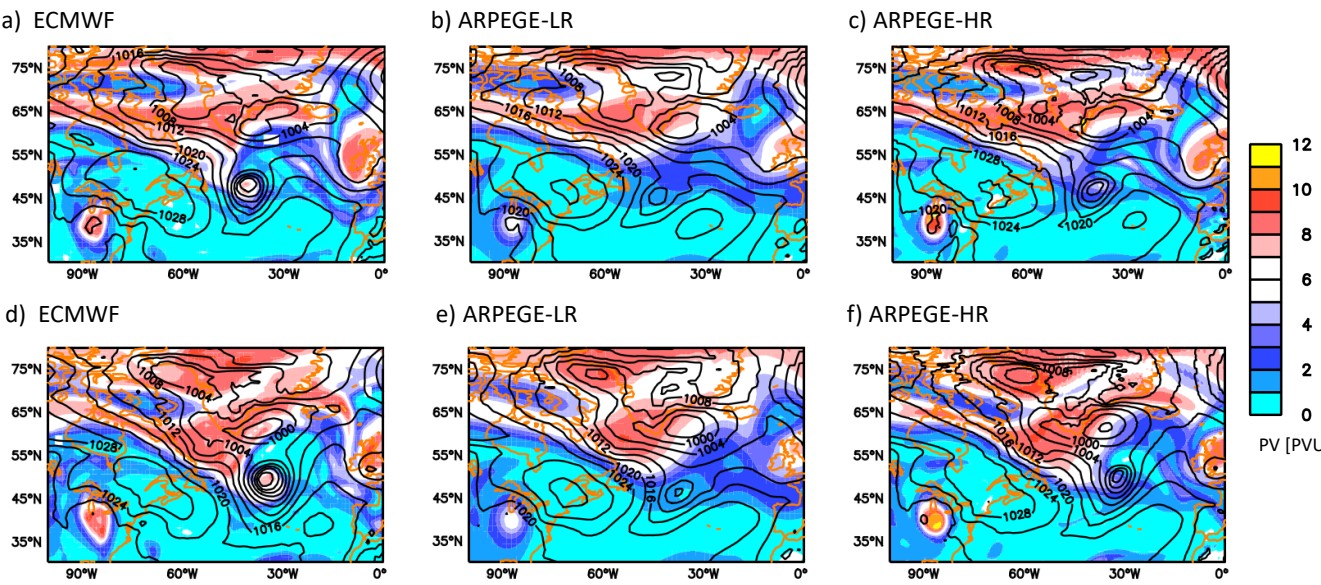

**Figure 8.** The 250 hPa PV (shaded) and mean sea level pressure (contoured) during the maximum deepening phase of the Stalactite cyclone. a–c) 00 UTC 1 October 2016, and d–f) 12 UTC 1 October 2016. a,d) ECMWF analysis; b,e) ARPEGE-LR hindcast;, and c,f) ARPEGE-HR hindcast. All hindcasts were initiated at 00 UTC 29 September 2016 and the colour scale applies to all plots. LMDZ-LR(-HR) plots at the same time are shown in Fig. S4.

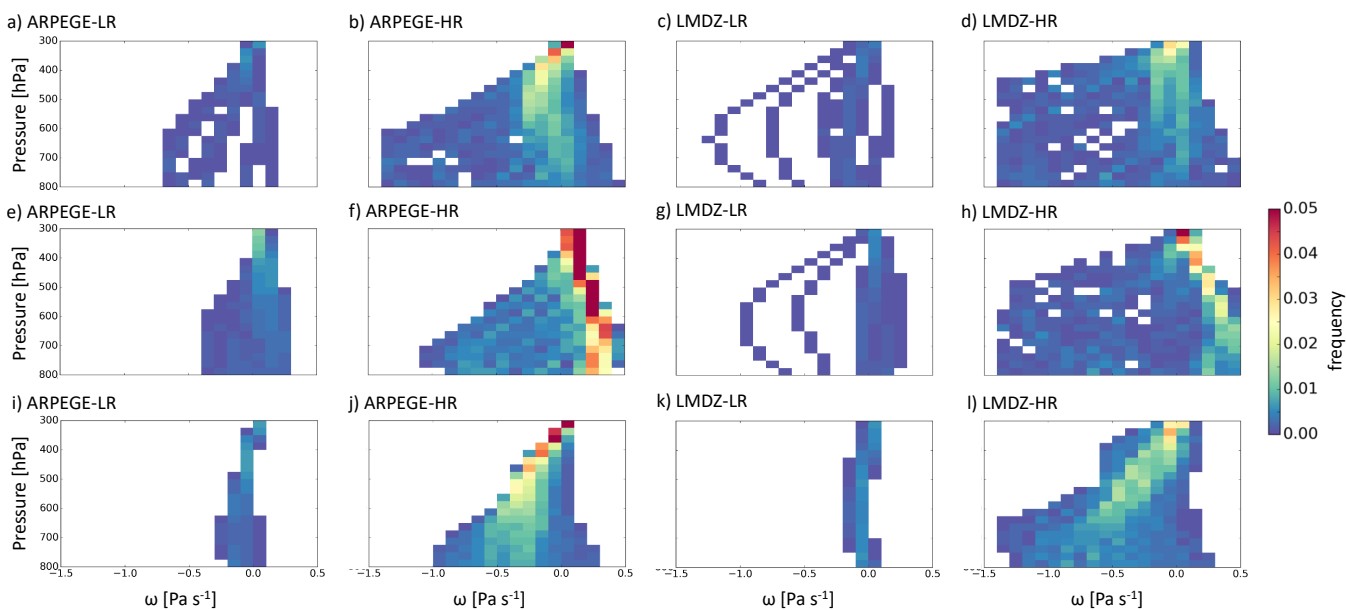

**Figure 9.** Bivariate histograms of vertical velocity *vs.* pressure averaged over a $3° \times 3°$ area centred on the cyclone during the mature stage of the cyclone around maximum depth (c. 12 UTC 2 October 2016, T + 33–36 h). For a–d) $\omega_{QG}$; e–h) $\omega_{diab}$; and i–l) $\omega_{dyn}$. For a,e,i) ARPEGE-LR; b,f,j) ARPEGE-HR; c,g,k) LMDZ-LR; and d,h,l) LMDZ-HR. The colour scale applies to all plots. The hindcasts were initated at 00 UTC 1 October 2016.

.





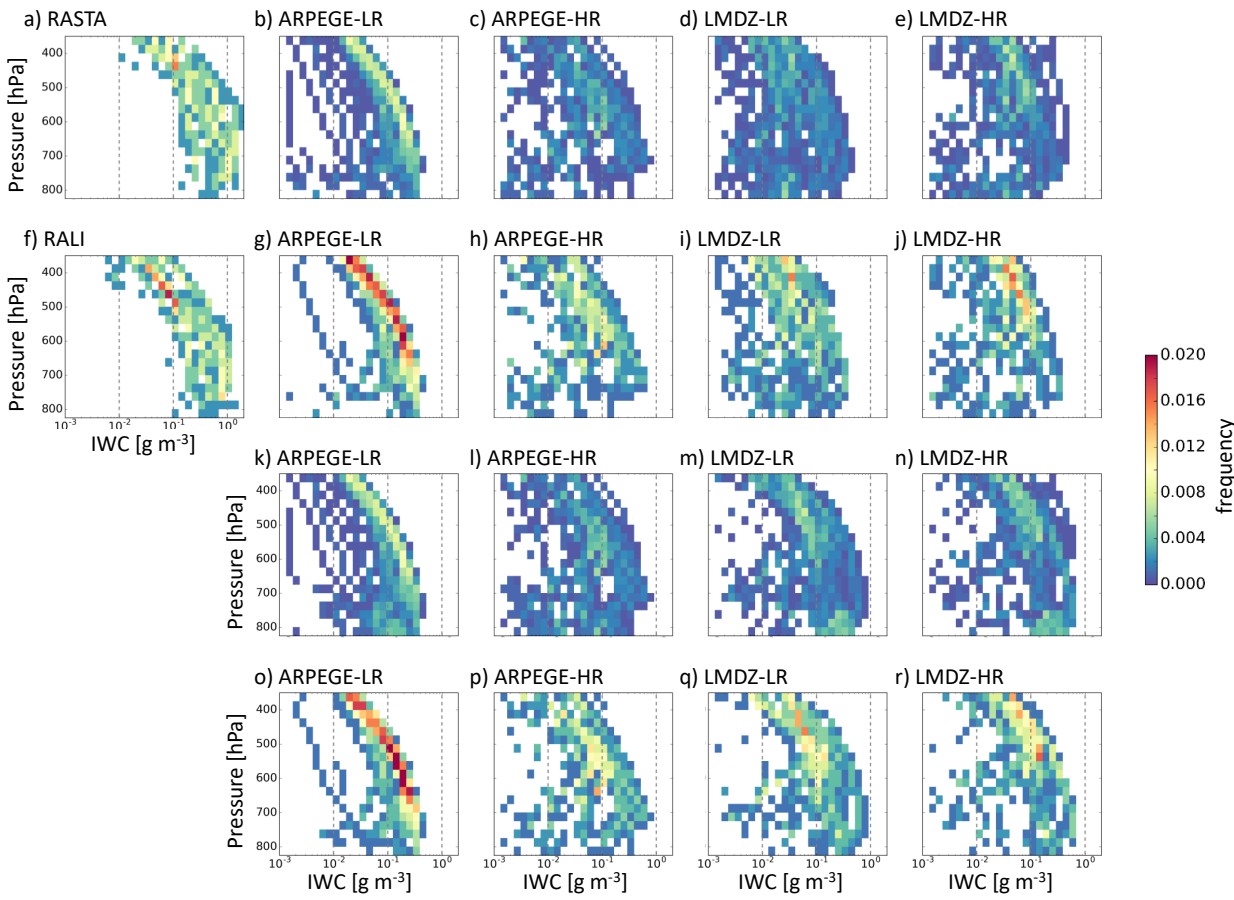

**Figure 10.** Bi-variate histograms of ice water content *vs.* pressure for F7 for a) RASTA observations (radar only); f) RALI (radar + lidar) observations; b–e) hindcast output using "potential" ice water content (cloud ice + snow) without applying a mask to the observations; g–j) hindcast output of "potential" ice water content (cloud ice + snow) and with the observation mask applied; k–n) hindcast for "maximum" ice water content (ice water content + liquid water content) without the observation mask applied; and o–r) hindcast of "maximum" ice water content (ice water content + liquid water content) with the observation mask applied. For b,g,k,o) ARPEGE-LR; c,h,l,p) ARPEGE-HR; d,i,m,q) LMDZ-LR; and e,j,n,r) LMDZ-HR. The hindcast data is initiated at 00 UTC 1 October 2016 and uses the nearest-gridpoint to the flight path from the two times surrounding the flight path (12 and 15 UTC 2 October 2016; T+36–39 h). The flight occurred from 1300–1600 UTC. The colour scale applies to all panels, and the histograms have been normalized by all points.

.



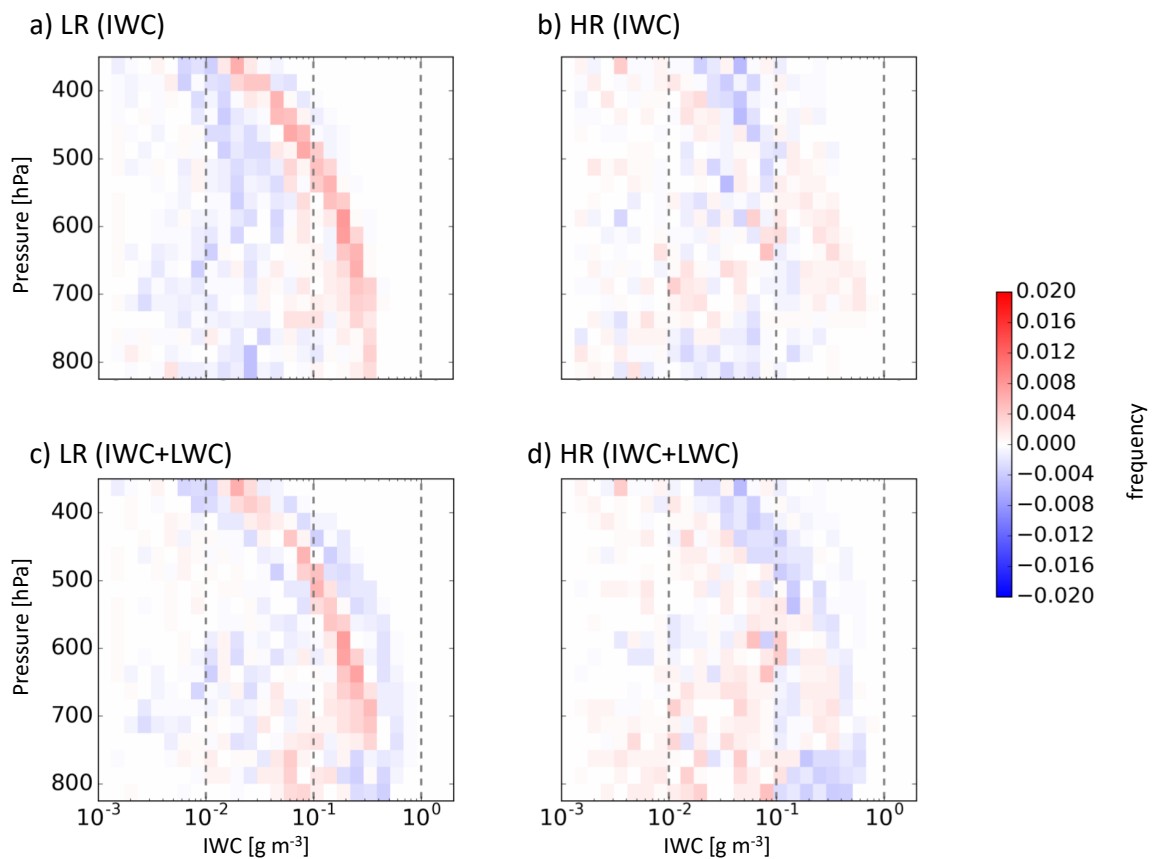

**Figure 11.** Difference bi-variate histograms for F7 of ice water content *vs.* pressure between ARPEGE and LMDZ for a) LR differences in "potential" ice water content (cloud ice + snow) only (Fig. 10b - Fig. 10d) ; b) HR differences in "potential" ice water content (cloud ice + snow) only (Fig. 10c - Fig. 10e); c) LR differences in "maximum" ice water content (ice water content + liquid water content) (Fig. 10k - Fig. 10m); and d) HR differences in "maximum" ice water content (ice water content + liquid water content) (Fig. 10l - Fig. 10n). Reds refer to ARPEGE having a larger quantity and blues for LMDZ. The colour scale applies to all panels. The hindcasts are initiated at 00 UTC 1 October 2016 and uses the nearest-gridpoint to the flight path from the two times surrounding the flight path (12 and 15 UTC 2 October 2016; T+36–39 h).

.





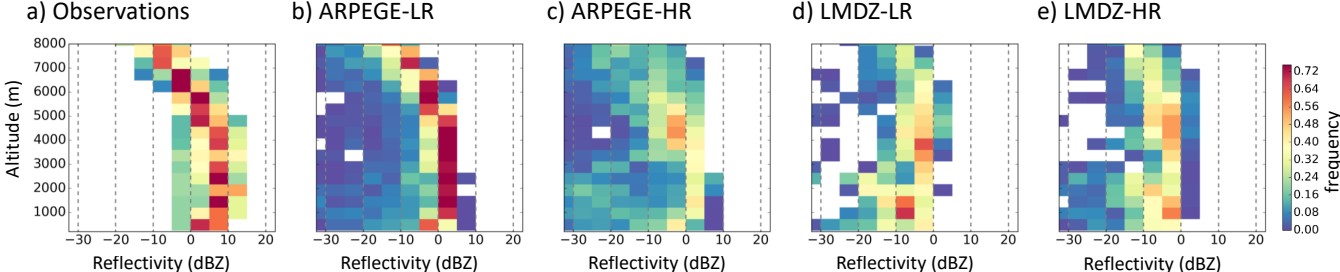

**Figure 12.** Contour Frequency Altitude Diagrams (CFADs) of Radar Reflectivity for F7 a) RASTA observations, b) ARPEGE-LR, c) ARPEGE-HR, d) LMDZ-LR, and e) LMDZ-HR. The hindcasts are initiated at 00 UTC 1 October 2016 and uses the nearest-gridpoint to the flight path from the two times surrounding the flight path (12 and 15 UTC 2 October 2016; T+36–39 h). The colour scale applies to all panels. No mask to the observations has been applied here.

.





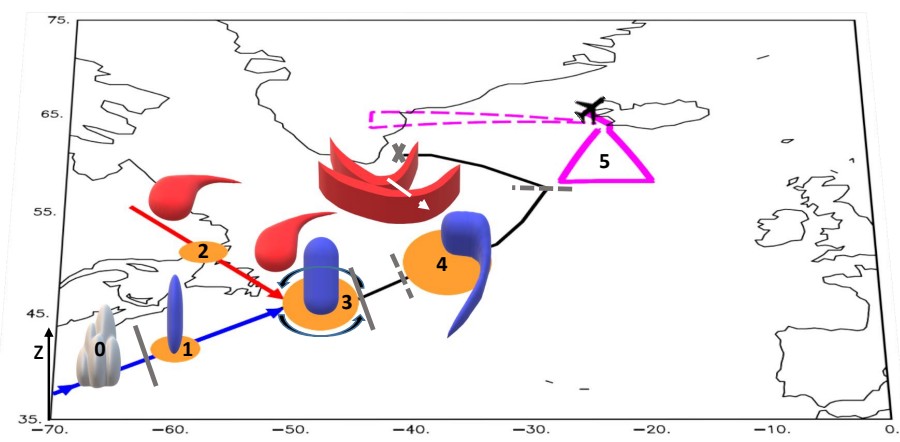

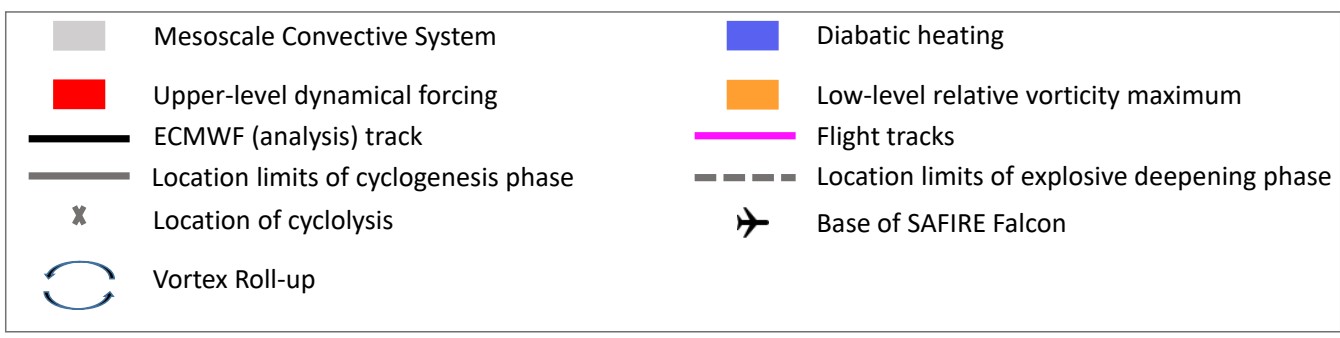

**Figure 13.** A schematic of the Stalactite Cyclone. 0) the Mesoscale Convective System that initiates the diabatic Rossby vortex (1) that travels along the blue arrow. The northern precursor (2) with upper-level PV cut-off that moves towards the diabatic Rossby vortex and initiations a roll-up between the two precursors at cyclogenesis to create the Stalactite Cyclone (3). Explosive deepening occurs as a result of strong diabatic heating throughout the column and the interaction with a series of embedded upper level high PV regions (4). Flight observations (5) indicate that ice water content is underestimated and so could have impacts on the diabatic heating and evolution of the cyclone.

.