# Peer review of "Representation by Two Climate Models of the Dynamical and Diabatic Processes Involved in the Development of an Explosively-Deepening Cyclone During NAWDEX"

_Weather and Climate Dynamics, 2020_

## Referee Comment (RC2) · Anonymous Referee #2 · 10 Nov 2020

My summary: This manuscript examines the ability of GCMs to capture extratropical cyclone development and intensification using a case-study. The manuscript considers two separate models and runs each at two resolutions. A diagnostic based on the quasi-geostrophic omega equation is utilized to separate dynamic and diabatic contributions to vertical motion. The model results are compared with observations.

Recommendation: I recommend accept conditional on minor revisions. I find the paper to be well written and relatively easy to read. I appreciate the new diagnostics and the comparison with observations. There are a few details about the technical set up of

analysis that need to be explained better (as line-by-line comments for details). There is an issue with the GQ omega that also needs more explanation.

Line 102: Possible verb tense disagreement for the word "interacts". I think you are talking about multiple PV anomalies, so it should be interact.

Line 104: For the footnote, there is a typo: the word is should follow the phi symbol.

Line 106: Placing a comma between trough and cyclonic would make the sentence easier to read.

Line 109-110: Wouldn't Gwendal say that the cyclone participated in the NAO regime transition rather than saying it "occurred during"? I say this in jest, I'm sure you will address this later on in the manuscript.

Line 153: You write: "For both models hindcasts are initiated at 0000 UTC on 27–29 September, and 1–2 October 2016 .."

I don't understand the logic of the choices of two separate initialization times? Or to put it differently, why is there no initiation at 00 UTC on Sept 30? Also, those initiated on Oct 2 will have already had the genesis in the IC, I think. So what are they used for?

Also, you state on line 162 that hindcasts initiatied on Sep 27 and 28 at low resolution did not produce the cyclone. So I'm confused as to why these dates are included in the sentence on line 153. I acknowledge that I may be misunderstanding something you've said.

Line 212: Typo on the word ClouSat - missing the d.

Line 231: How interesting that the time-average of the static stability works in this calculation.

Line 247: What are the horizontal boundary in a global model? I suppose you choose some appropriate distance?

Line 255: You write: "Inversion of the two previous equations allows to separate " Sounds a bit awkward. Maybe there is a word missing between allows and to?

Line 257: What do you mean by "Vertical velocity intervenes . . .." I don't follow what you mean by intervenes in this context.

Line 286: This 24-hour delay in deepening seems to run contrary to your statement on Line 217, where you wrote:

"In reality, for the hindcast that is compared against the observations in this study (initiated at 00 UTC 1 October 2016) neither timing nor positioning adjustments are required. "

I assume that this delay in the intensification that you discuss on Line 286 is in a hindcast initialized earlier than the Oct 1 date, but it would be good to include more clarity about this, either here, or in/near the sentence on Line 217.

Line 319-330. One thing that the table makes clear, that I don't think is mentioned: the HR models generate a stronger DRV than ECMWF. This should be stated in the text I think.

Line 325: Figure 4 is a something of a proof of concept for your dynamic/diabatic separation. Though it is might be worthwhile to mention that the dynamics and diabatic component are tightly coupled.

Line 332: Section 4.2.2. How interesting that the LR model is more different from the HR model for the upper-level disturbance. We tend to think of diabatic forcing related to moisture parameterizations being the issue, but here that is not the case. Does this suggest that region in which the dynamics and diabatic components must interact are more sensitive to the role of resolution? Or is this an initial value problem? Or is this unique to this case-study. Alas, that is not something you can easily answer I imagine. Interesting to think – no need to provide a detailed response to this comment however.

Line 359: Section 4.2.3 Regarding the statement about these biases being the cause

for the changes in track path, are you sure you can attribute it to this mechanism? If so, how?

Line 375: You write: "The averaged quasi-geostrophic baroclinic conversion is roughly reduced by two thirds in magnitude compared to that directly calculated from the model $\omega$ but is consistent . . ."

That is quite a reduction. This is the only thing that I've read (so far) in this manuscript that makes me pause and wonder, should we be so confident in their diagnostics? There is not much discussion. Are you hoping to ignore it so that others do as well, or is there good reason to be unbothered by this difference? What is the difference between the Q-G baroclinic conversion and that using modeled omega for the cyclogenesis phase?

Line 415: Figure 9 What do these plots look like for modeled omega?

Line 487: Figure 12 is a bit disheartening isn't it? As you say, ARPEGE-LR looks the most like the observations. But the HR models more closely match the dynamics of reanalysis. So what does this mean? Does the CFAD have no relationship with the dynamics of the cyclone? Or should we raise more questions about the dynamics in reanalysis? It is a tough figure to interpret, but I am glad that you include it in the manuscript.

---

## Author Comment (AC1) · 10 Dec 2020

**Response to Reviewers for WCD-2020-43**

*Dear Editor,*

*We thank both reviewers for taking the time to review and comment upon our manuscript. Following the comments we have made substantial revisions to the manuscript.*

*The main change is a significant reduction in the length of the manuscript, which now makes it more in line with other, published, articles in this journal.*

*We have also provided an in-depth response to the inversion of the QG equation (with some addition in the manuscript) that we hope alleviates the reviewers' concern. Some figures are inserted in the reply document showing the strong similarities between the inverted and model vertical velocities even though some differences in the amplitudes do exist. We do not expect inversions of the QG equation to entirely match up to values modelled in full GCMs but the strong resemblance between the two fields together with the reasonable amplitude of the retrieved field give confidence on the results.*

*We provide detailed responses below with our comments and changes in green italics.*

*Kind regards,*

*David Flack, Gwendal Riviere, Ionela Musat, Romain Roehrig, Sandrine Bony, Julien Delanoe, Quitterie Cazenave and Jacques Pelon*

**Referees' comments are in black and authors' answers in green**

**Response to Anonymous Referee #1**

I have more expertise in the dynamics of extratropical cyclones than in the cloud microphysics. Thus, my comments mostly deal with the general presentation of the study and that part of the content which is related to the dynamics of the system. General comments: In this study, the authors evaluate the performance of two climate models simulating the evolution of one North Atlantic extra-tropical cyclone (so called Stalactite Cyclone). The authors compare the models with two horizontal resolutions against ECMWF operative analyses. Quasi-geostrophic omega equation is applied to distinguish the dynamic and thermodynamic (diabatic) processes within the cyclone. Furthermore, airborne measurements of microphysical properties of the clouds are compared to the model simulations. I found the topic of this paper interesting and suitable for the scope of WCD journal. The language is clear and understandable. However, I think that the paper suffers from some flaws in its organization, which are mostly due to its length. Therefore, more effort is needed to make the presentation of the paper and its message clearer for the readers. I hope that my comments will help the authors in this work.

Recommendation: major revision

Major comments, which are partly connected to each other:

1. In general, I think that the authors are trying to put slightly too much content into one paper and thus the manuscript was partially hard to read and lacking a clear and consistent storyline. In my opinion, answering comprehensively to the objectives 1 and 2 listed in the introduction would suffice perfectly to the topic of this paper. In its current form the point 3 feels somewhat disjoint and was, to my opinion, particularly difficult to understand. I think that the main results from points 1 and 2 (e.g. the fact that there was no change in the relative importance of diabatic processes with increased resolution) are very interesting and worth publishing alone without focusing on the comparison of microphysical properties with the models and observations, which in turn would be suitable as its own manuscript if investigated properly. Moreover, there is room for improvements in the organisation of the paper. For instance, between the introduction and Section 3, the authors present the overview of the Stalactite Cyclone. I think that this kind of section would better belong right before Section 4, where the representation of the cyclone by the models is presented. Now, after reading all the lengthy details related to the models, observations and the equations, the reader has already forgotten the whole overview of the cyclone itself.

*We have extensively edited the manuscript from the original submission to reduce its length, in particular in the methodology section. We have re-read and understood your point about point 3 and as such we have moved question 3 to question 4 and introduced a new question to act as an intermediary between points 1 and 2 and the final question. We feel that this new question puts greater emphasis on the link between the model and the observations, and why we look at microphysical properties.*

*The new question introduced reads as follows:*

*"Are there any differences in the diabatic processes related to microphysical properties between the different models?"*

*We feel this question improves the link as first of all it determines if there are differences in the diabatic heating (previously discussed) and links them to the observations by considering diabatic heating related to microphysics in the models. Therefore we feel there is greater justification and link between Section 4.3 and the other parts of the paper. Further discussions on the importance of the last point are included in the response to the next comment.*

*Concerning the comment related to changing the order of section 2 and section 3, we decided not to change the order of these sections because section 3 dedicated to the presentation of the models, diagnostics and observation needs to be read after the synoptic description of the cyclone. This is needed to understand the choice of the initial time of the simulations in section 3.1 and to visualize the flight legs within the cyclone in section 3.2. As previously mentioned, we reduced the length of section 3 (from 5 pages to 3 pages), in particular section 3.3 dedicated to the presentation of the omega equation. As such, the description of the main features of the cyclone in the models (cyclone track and pressure evolution in the beginning of section 4) is not so far from the overall description of the cyclone (section 2).*

2. Related to comment 1, the study as a whole is a very long read. The paper in its current form has > 11 000 words (from Abstract to Acknowledgements), 13 figures and altogether 38 pages (+ additional seven pages in the supplementary material). Although to my knowledge WCD does not explicitly limit their manuscript lengths, the longevity of the paper did make me lose my interest in reading through it at the first time. For reference, AGU journals (https://www.agu.org/Publish-with-AGU/Publish/AuthorResources/Text-requirements) are recommending up to 25 publication units (PU) for their manuscripts, where 1 PU is 500 words or 1 display element (figure or table). With the 13 figures and 2 tables, the current manuscript would be left with 15 PUs for the main text, which would correspond to 7500 words. The current length is now > 50 % longer. Please consider condensing the paper. I think the best way to do this would be to focus clearly only on objectives 1 and 2 (as suggested in comment 1), or expressing really the main results from points 1, 2 and 3 in a much more condensed way.

*Using word count software indicates that the original manuscript's word count is 9896 words, and so we base our reductions on this figure.*

*We appreciate that this is a long manuscript and as such we have endeavoured to reduce the length of the manuscript. We have decided to take the route of shortening whilst still discussing the three points. We chose this option as we feel that section 3 is important to be included within this manuscript, and is important work for the community as a whole, as it brings forth some concerning aspects of model behaviour, it indicates areas that could be improved, and areas that can have important feedbacks on the interpretation of cyclones within the models. Part of the implication of Section 3 is that we may have a similar cyclone in both climate models but we are getting it for different reasons and this is feeding back onto the dynamics with the different deepening rates. Thus, Section 3 helps to explain differences seen in Section 2. Indeed in the Summary of the manuscript we state "Finally,*

*and arguably most critically, it warns that although climate models may produce similar cyclones they can be doing so for very different reasons and these reasons are likely to have an influence upon other areas of the climate system and the response of model cyclones to climate change.''* Further to this, Reviewer 2 has also indicated that the inclusion of these remaining figures is important to include within the manuscript. Thus we feel these are good justifications for keeping the discussion around point 3.

*The main areas we have focused on is reducing the methodology section (including transferring the model descriptions into a table and reducing the diagnostics part), but also condensing the main results. After our reductions the word count is 7826 [which reduces our submitted manuscript page count from 38 to 30 pages: 25 of those as text pages (including data availability, the link to supplementary material, author contribution and acknowledgements)]. We think this significant reduction of words (2163 less) and page count makes the paper more readable as requested by the reviewer while keeping its scope the same.*

3. The manuscript is full of acronyms which are mostly related to model names. I think that the abundant use of acronyms was one of the reasons which made my reading less enjoyable. In the abstract, you call the models with names CNRM-CM6-1 and IPSL-CM6A. In the main text, these models are referred mainly to with their atmospheric components (if I understood correctly) called LMDZ and ARPEGE. Again in the summary you change to CNRM-CM6-1 and IPSL-CM6A. This was very confusing for a reader who is not familiar with these models. The inclusion of LR or HR in some places makes the names even longer (e.g. IPSL-CM6A-LR(-HR)). I strongly suggest to use short and consistent names throughout the whole manuscript, e.g. CNRM and IPSL, with possible -LR and -HR suffixes.

*We appreciate that the over use of acronyms can make reading the manuscript difficult, and that our change of acronyms may not have been as clear as we intended. However, we also note that when comparing two models we need to refer to them by their full acronyms first and shorter ones throughout the main text so this will naturally lead to a large amount of acronyms being used.*

*We have tried our best to reduce the number of acronyms used throughout the paper. To facilitate this we now only introduce the full length, official, climate model names in the abstract and methodology when the models are introduced. Throughout the rest of the paper we only consider the (shorter) atmospheric component model names. We use the full length atmospheric component names in the abstract, the introduction of the models and briefly again in the summary to act as a reminder for the contracted acronyms we use throughout the main manuscript. Thus now we refer to LMDZ and ARPEGE throughout the manuscript (with the suffixes LR and HR used only where necessary). We have also improved the signposting of this within the main manuscript.*

Minor comments

1. Title: I have a feeling that the part of the title "How Well do Models Represent the Development of Extra-Tropical Cyclones?" is a bit too vague, given that you have investigated only two models and one cyclone. Please consider having a more specific title.

*The new title is now: "Representation by Two Climate Models of the Dynamical and Diabatic Processes Involved in the Development of an Explosively-Deepening Cyclone During NAWDEX".*

2. 'L4 and thereafter: CMIP5 and CMIP6 should be written together, and not "CMIP 5" or "CMIP 6" (see e.g. https://pcmdi.llnl.gov/CMIP6/).

*This has now been corrected throughout the manuscript.*

3. L4: Can you write down both resolutions explicitly, or leave both away? Writing only one resolution with its numerical value (0.5, HR) left me missing the other one, because at least I am not aware of the CMIP6 native resolution (LR).

*We had originally put it as CMIP6 and specified the value as the two different models had different CMIP6 resolutions (and so we did not want to increase the length of the abstract). However, we understand your point and have now referred to both resolutions within the manuscript (as it is important information to include in the manuscript). We have expressed the resolutions by giving approximate grid lengths in km (150-200 km for LR and c. 50 km for HR).*

4. L84: Section 2 title "NAWDEX IOP 6: The Stalactite Cyclone" could be more descriptive, e.g. The development of the Stalactite Cyclone, or The life cycle of the Stalactite Cyclone.

*This has been re-named to ''The Lifecycle of the Stalactite Cyclone (NAWDEX IOP 6)''.*

5. L89: I was confused about the word Diabatic Rossby Vortex (DRV), because Boettcher and Wernli (2013) talk about Diabatic Rossby Waves (DRW). If they mean identical phenomena, please consider adding a clarification where you indicate that they mean the same.

*There is a long-standing debate in the literature about whether or not to call the phenomenon being referred to a Diabatic Rossby Vortex or a Diabatic Rossby Wave (see the Appendix of Boettcher and Wernli (2013) for more on this debate). In the literature it is understood that both of these terms refer to the same phenomenon. We use the term Diabatic Rossby Vortex within the manuscript as we feel this term better suits what we see, i.e. a single vortex moving up the eastern seaboard of the USA (rather than a wave). We do not wish to have (or get into) a debate on which term should be used. As suggested we have added clarification (by means of a footnote) for readers to indicate they are identical phenomena so as not to confuse readers that are either not used to the term or the phenomenon.*

*The exact wording of the footnote is as follows*
*''In essence this is the same phenomenon as a diabatic Rossby wave (see Appendix of Boettcher and Wernli, 2013).''*

6. L124: Why do you use ECMWF operational analyses for comparison, and not ERA5 reanalysis? Isn't ERA5 considered more reliable because more observations have been

assimilated into it? However, if you have strong rationale to use ECMWF analyses, please include it in the paper.

*We have used ECMWF operational analysis to keep consistency with the initialization of our model simulations and to determine whether any initial shock had occurred. A brief sentence along these lines has now been added to the start of section 3.*

*The end of the sentence now reads "as a consistent baseline with the analysis state."*

7. L243: It was a bit unclear for me why you split the diabatic heating into components here, because in the results (Sect. 4.4.1) you only talk about the omega/baroclinic conversion due to diabatics and not these single components.

*This is a hangover from an older version that we had missed, so thank you for indicating this to us. This has now been removed as part of the requested reduction in the length of the manuscript .*

*In the older version we had originally decomposed the diabatic heating term into its components to consider which processes / parameterizations were providing different heating rates in the models.*

8. L245: Isn't latent heating including also the freezing and melting of ice droplets in the clouds, but you mention only condensation and evaporation. Why?

*We agree that latent heating would also include the freezing of cloud and rain water droplets and melting of ice crystals/snow. The condensation and evaporation were mentioned as examples (and it was not meant to be an exhaustive list) as these are tendencies explicitly produced by the model (the latent heating was included just to give a reference meaning to the phrase large-scale condensational heating and evaporation - it was not meant as an exhaustive summary of what latent heating was). This has now been removed in relation to the comment above and the need to reduce the length of the manuscript.*

9. L248: You mention that on average most of the modelled vertical motion is recovered using QG method. However, in Figure 7, there are some quite large discrepancies between Model and Inverted baroclinic conversion. How confident are you with the contributions of dynamic and diabatic processes if their sum (as indicated by Eq. 5) does not match well with the modelled baroclinic conversion? Did you verify how well the diagnosed omega (from QG equation) matches with the omega from the climate models? If yes, please consider adding a couple of sentences about the verification.

*Given that we are using a full atmospheric model we do not expect the inverted QG omega to be equal to the modelled QG omega. However, we expect the QG assumption to be enough to diagnose the main sources of the cyclone development at synoptic scales.*

*The fact that the timing of the evolution of the inverted omega matches that of the modelled omega in Figure 7 is one reason mentioned in the original submission to provide confidence in our inversion. The other main reason relies on the strong similarities between the inverted and modelled omega (compare the contours and shadings in each panel Fig. R1 of the*

*present document). The spatial resemblance is striking for both high and low resolution runs. For the low resolution runs and In some regions, the modelled omega is slightly underestimated by the inverted omega. For instance, the peak values of the inverted omega is roughly two thirds of those of the modelled omega in Fig.R1a,b. For the high resolution runs, the modelled and inverted omega get roughly the same amplitude (see also Fig.R3 of the present document) but it varies from case to case. This information will be provided in the revised version of the paper. An additional point which makes us confident in our decomposition is when we look at the effects of the two components on the total (see Fig. R2 and our reply to reviewer 2's comment). The dynamical and diabatic vertical velocities are not necessarily co-located (Fig. R2b,c) and this is only the sum of the two components that matches the model omega. It means that both components are crucial to recover the full signal.*

[Figure]

*Figure R1: Inverted (QG; shaded; int: 0.1 Pa/s) and model (contoured; int: 0.1 Pa/s) vertical velocities during cyclogenesis on 30 Sep 00 UTC (a,c) and mature stages on 2 Oct 12 UTC (b,d) of the Stalactite Cyclone. a,b) are for ARPEGE-LR and c,d) for ARPEGE-HR. Similar results are obtained for LMDZ at both resolutions (not shown).*

10. L252: Why do you express the omega equation twice? I think this is unnecessary and removing Eq. 2 would shorten your manuscript.

*The purpose of Eq. 2 was to make the splitting of the equation into dynamic and diabatic components explicit. However, such a decomposition can be said in words and because the manuscript needed to be shortened, Eqs. (2) and (3) have been suppressed.*

11. L416: How can you say that the larger ascents mainly arise from diabatic processes? For me it's very difficult to conclude that based on only the figure. Do you have some quantitative analysis behind your statement? In any case, it would be interesting to see some quantitative values related to how much the increased resolution increases the omega due to diabatic heating and the omega due to dynamics. For example, something similar as in Table 2 in Sinclair et al. (2020) (https://wcd.copernicus.org/articles/1/1/2020/)

*Figure 9 shows bivariate histograms of vertical velocities as a function of vertical levels. A new version is available in Fig.R3 of the present document highlighting more the probabilities of extreme values. In addition, the figure slightly differs from the submitted version because a spatial filter applied to the Q-vector remained in our code that appears to be useless and tended to underestimate the dynamical omega. Considering this correction, Fig.R3 led to the following conclusions. For ARPEGE, the peak values of the dynamical and diabatic omega are roughly the same while for LMDZ the diabatic omega reaches higher maxima than the dynamical omega. It shows that LMDZ has a stronger diabatic rate than ARPEGE for both resolutions and this provides an explanation for the stronger deepening rate in the simulations of the former model than in the latter. So the sentence will be changed in the revised version. Also, in the revised paper, all figures that use inverted omega are updated to remove the spatial filter from the inversion. Such a correction leads to slightly stronger dynamical vertical velocity but this does not change the main results.*

12. Fig 1a, 2a and 7: please could you add horizontal (and vertical) grid lines so that the days are easier distinguishable from each other.

*We feel that gridlines clutter up the plots. Instead we have added tick marks on the axis to aid in interpretation and so that the days are more easily distinguishable. The date is marked at 12 UTC on each day, we have also made these additions to the corresponding supplementary material figures as well.*

13. Fig 1a: in addition to the grid lines perhaps you could add shadings and small labels to the graph so that initiation and deepening phases are easier visible? And to my eye, the deepest phase of the cyclone seems to be about 6 hours earlier than what is indicated with the dotted line.

*We have introduced shadings to indicate the regions to indicate the different phases, we have also changed the deepest phase of the cyclone to a region covering both times, and added the corresponding labels to the figure.*

14. Fig 1d: The labels in the thickest black contours are not seen properly. Furthermore, there seems to be some gaps in the contours in the middle of the domain.

*The figure has been re-plotted to remove the gaps in the middle of the domain, and also remove the labelling of the bold contours (we feel that we can remove these labels as the other labels (and caption) are sufficient to give an idea of the spacing.*

15. Fig 1e: The colorbar and its labels are too small.

*The colour bar has now been increased in size.*

16. Fig 9: Can you add the titles (Total, omega_diab and omega_dyn) to the plots. It would be easier to interpret the plot when you don't need to check from the caption what the different rows express.

*Titles have now been added to this figure.*

Finally, it would help the reviewers if the figures were included within the main text and not at the end of the manuscript.

*In the original latex file the figures were in the correct place in the document but due to the size of figure 1 all figures were produced at the end of the manuscript. We have spent time correcting this and reducing the size of figure 1 to ensure it is now in the correct place and will allow all of the other figures to be in the correct place in the manuscript to alleviate this concern.*

**Response to Anonymous Referee #2**

My summary: This manuscript examines the ability of GCMs to capture extratropical cyclone development and intensification using a case-study. The manuscript considers two separate models and runs each at two resolutions. A diagnostic based on the quasi-geostrophic omega equation is utilized to separate dynamic and diabatic contributions to vertical motion. The model results are compared with observations.

Recommendation: I recommend accept conditional on minor revisions. I find the paper to be well written and relatively easy to read. I appreciate the new diagnostics and the comparison with observations. There are a few details about the technical set up of analysis that need to be explained better (as line-by-line comments for details). There is an issue with the GQ omega that also needs more explanation.

Line 102: Possible verb tense disagreement for the word "interacts". I think you are talking about multiple PV anomalies, so it should be interact.

*Thank you, this has now been changed.*

Line 104: For the footnote, there is a typo: the word is should follow the phi symbol.

*Thank you, this has now been changed.*

Line 106: Placing a comma between trough and cyclonic would make the sentence easier to read.

*Thank you, this has now been changed.*

Line 109-110: Wouldn't Gwendal say that the cyclone participated in the NAO regime transition rather than saying it "occurred during"? I say this in jest, I'm sure you will address this later on in the manuscript.

*This has now been changed by "participated in" because the ridge building ahead of the cyclone initiates the blocking and this is supported by Maddison et al (2019)'s findings.*

Line 153: You write: "For both models hindcasts are initiated at 0000 UTC on 27–29 September, and 1–2 October 2016 .."

I don't understand the logic of the choices of two separate initialization times? Or to put it differently, why is there no initiation at 00 UTC on Sept 30? Also, those initiated on Oct 2 will have already had the genesis in the IC, I think. So what are they used for?

Also, you state on line 162 that hindcasts initiated on Sep 27 and 28 at low resolution did not produce the cyclone. So I'm confused as to why these dates are included in the sentence on line 153. I acknowledge that I may be misunderstanding something you've said.

*This section has been substantially edited in the revised manuscript to take into account the comment of reviewer 1 feeling the paper was too long. Thus in the new manuscript we focus*

*on the two runs considered throughout the manuscript (29 September and 1 October). The original intention of this section was to justify the simulations considered, however we can see why this would lead to a lack of clarity.*

*The first initiation time (29 Sep) is used to examine the entire life cycle of the Stalactite Cyclone to answer the following question: Are the models able to reproduce the life cycle of the cyclone once all precursors are present in the analysis ? For simulations starting at earlier times (27, 28 Sep) the LR resolutions did not produce a cyclone (due to a lack of DRV and upper-level PV interaction) and the comparison between LR and HR simulations was useless. The second initiation time (2 Oct) is used for observational comparisons to ensure similar cyclone structure and position to reality and to focus on differences in cloud microphysics.*

Line 212: Typo on the word ClouSat - missing the d.

*Thank you for spotting this, this is in a section of the paper that has since been removed.*

Line 231: How interesting that the time-average of the static stability works in this calculation.

*In QG theory the static stability appearing in the omega equation should be dependent upon the vertical only. This is the reason why a time average has been applied in addition to a horizontal average. Note that a time dependent static stability would not change much the results because the horizontal average made over the Atlantic area tends to smooth the field already.*

Line 247: What are the horizontal boundary in a global model? I suppose you choose some appropriate distance?

*The horizontal boundaries are the boundaries used for the inversion (so as not to invert over the whole globe) and are restricted to a region where the QG assumption makes the most sense (i.e. the mid-latitudes). The inversion box is now defined in the manuscript as 70W to 0E and 35 N to 75N (this is the area depicted in Fig. 1b). This encompasses the entire lifecycle of the cyclone and ensures that the centre of the cyclone is far enough away from the boundary throughout its lifecycle, when QG most likely applies) to show no/limited boundary effects.*

Line 255: You write: "Inversion of the two previous equations allows to separate " Sounds a bit awkward. Maybe there is a word missing between allows and to?

*Yes, you are quite right there was a missing word. As part of the reduction of the manuscript based on comments from the other reviewer we have removed this section.*

Line 257: What do you mean by "Vertical velocity intervenes . . .." I don't follow what you mean by intervenes in this context.

*This has been re-worded for clarity, we meant occurs in.*

Line 286: This 24-hour delay in deepening seems to run contrary to your statement on Line 217, where you wrote:

"In reality, for the hindcast that is compared against the observations in this study (initiated at 00 UTC 1 October 2016) neither timing nor positioning adjustments are required. "

I assume that this delay in the intensification that you discuss on Line 286 is in a hindcast initialized earlier than the Oct 1 date, but it would be good to include more clarity about this, either here, or in/near the sentence on Line 217.

*Yes, you are correct in assuming the two statements refer to runs initialized at two different times. When we compare the observations we are using runs initiated on 1 October. We did mention this in the Methodology, however we have now made this clearer, as we appreciate it could have been easily missed. We highlight again at the start of Section 4.3 that we have changed runs to only look at the run initiated on 1 October.*

Line 319-330. One thing that the table makes clear, that I don't think is mentioned: the HR models generate a stronger DRV than ECMWF. This should be stated in the text I think.

*You are quite correct in saying that the DRVs in the HR models are stronger than ECMWF. We did not mention this in the text as we were focusing on whether the LR models could create a DRV - however, we do now mention this briefly in the text as suggested.*

Line 325: Figure 4 is a something of a proof of concept for your dynamic/diabatic separation. Though it is might be worthwhile to mention that the dynamics and diabatic component are tightly coupled.

*We now mention that the two components are tightly coupled.*

Line 332: Section 4.2.2. How interesting that the LR model is more different from the HR model for the upper-level disturbance. We tend to think of diabatic forcing related to moisture parameterizations being the issue, but here that is not the case. Does this suggest that region in which the dynamics and diabatic components must interact are more sensitive to the role of resolution? Or is this an initial value problem? Or is this unique to this case-study. Alas, that is not something you can easily answer I imagine. Interesting to think – no need to provide a detailed response to this comment however.

*Yes, we felt it was an interesting result too. We would say it certainly merits more investigation but would agree that there are sensitivities to the role of resolution here and potentially an initial value problem too given later runs see an improvement in the simulations. We are unsure if this is unique to this case study, but we are aware that similar (albeit not exactly the same) results occur with the subsequent cyclone that followed (which was targeted for future work). In this other case the interactions and balance between the different components of baroclinic conversion were slightly different so it is a little harder to answer.*

Line 359: Section 4.2.3 Regarding the statement about these biases being the cause for the changes in track path, are you sure you can attribute it to this mechanism? If so, how?

*We have weakened the language used within this section, saying it is "likely". However, we do add a little more explanation to this comment as well. We note that it is straight after cyclogenesis that the track starts to deviate in the forecast when the cyclone tracks are discussed in Section 4.1. This can be seen by comparing the tracks of the HR and LR runs. Given that the merger is more apparent in HR runs compared to LR runs it is strongly suggestive that this is the cause. Another argument is based on previous studies on cyclone tracks (Gilet et al. 2009; Oruba et al. 2012, 2013; Coronel et al. 2015). The poleward motion of midlatitude cyclones occurs during baroclinic interaction of surface cyclones with upstream upper-level troughs. The presence of the upper-level trough tends to create poleward winds near the surface cyclone center and this advects the surface cyclone perpendicularly to the mean jet axis. In the present study, the LR simulations miss the initial interaction with the small-scale upper-level disturbance and this prevents poleward advection by the latter anomaly. Baroclinic interaction with upper levels happens later in the LR runs and this likely explains the delayed poleward motion of the surface cyclone in those runs.*

Line 375: You write: "The averaged quasi-geostrophic baroclinic conversion is roughly reduced by two thirds in magnitude compared to that directly calculated from the model ω but is consistent . . ."

That is quite a reduction. This is the only thing that I've read (so far) in this manuscript that makes me pause and wonder, should we be so confident in their diagnostics? There is not much discussion. Are you hoping to ignore it so that others do as well, or is there good reason to be unbothered by this difference? What is the difference between the Q-G baroclinic conversion and that using modeled omega for the cyclogenesis phase?

*We repeat our response to a similar comment from reviewer 1. This response gives reasons why we are unbothered by this difference - we were not trying to ignore it in the manuscript. In fact, we do address this within the manuscript and say why we can be confident in the results.*

*As already mentioned in the submission, the timing of the evolution of the inverted omega matches that of the modelled omega in Figure 7. The other main reason relies on the strong similarities between the inverted and modelled omega (compare the contours and shadings in each panel of Fig. R1 of the present document and Fig. R2.(a)). This is shown for the two models and is valid for both high and low resolution runs. Additionally, Fig. R2 shows that the dynamical and diabatic vertical velocities are not necessarily co-located (panels b and c) and this is only the sum of the two components that matches the model omega. These maps thus provide confidence in our decomposition. There is some underestimation of the amplitude of the modelled omega by the inverted omega in some regions of the low-resolution runs (see Fig.R1 of the present document and the associated reply to reviewer1's comment) but this is not seen in high resolution runs (see Fig. R3 of the present document). This information is provided in the revised version of the paper.*
*Note that we have made a correction in our code since the initial submission. A spatial filter was initially applied to the Q-vector (at that time we did not have the Inverse of the Laplacian*

*and we wanted to anticipate the effect of such an operator). This spatial filter remained in our code but was useless and tended to underestimate the dynamical omega. In the revised paper, all figures that use inverted omega are updated to remove the spatial filter from the inversion. Such a correction leads to slightly stronger dynamical vertical velocity. The main results are not changed but the ratio of ⅔ is not valid anymore. In fact, the inverted omega is now even closer to the model one than before.*

[Figure]

*Figure R2: Inverted (QG; shaded) and model (contoured) vertical velocities in LMDZ-LR run starting on 1 Oct 2016 for (a) the total inverted, i.e the sum of the dynamical and diabatic components, (b) the dynamical component and (c) the diabatic component.*

Line 415: Figure 9 What do these plots look like for modeled omega?

*Figure R3 represents Figure 9 with the modeled omega histograms included. The shapes of the inverted and model omega histograms are rather similar. More discrepancies appear for LMDZ-LR where we see that the peak values of the model omega are not reproduced by the inverted omega. These values correspond to grid points near 28°W and 58°N in Fig. R2(a). Despite this underestimation of the amplitude, the location of the peak values is well reproduced by the inverted omega.*

[Figure]

*Figure R3:* Bivariate histograms of vertical velocity vs pressure averaged over a 6°x6° longitude x latitude area centred on the cyclone during the mature stage of the cyclone around maximum deepening for (first row) model omega, (second row) inverted omega, (third row) dynamical component of the inverted omega and (fourth row) diabatic component of the inverted omega. (1st column) ARPEGE-LR, (2nd column) ARPEGE-HR, (3rd column) LMDZ-LR, (4th column) LMDZ-HR.

*Note that the new figure 9 (Fig. R3) is slightly different from the submitted version because of the removed spatial filter we were using in the submitted version.*

Line 487: Figure 12 is a bit disheartening isn't it? As you say, ARPEGE-LR looks the most like the observations. But the HR models more closely match the dynamics of reanalysis. So what does this mean? Does the CFAD have no relationship with the dynamics of the cyclone? Or should we raise more questions about the dynamics in reanalysis? It is a tough figure to interpret, but I am glad that you include it in the manuscript.

*We would agree that Fig. 12 is a bit disheartening, but we are pleased that you feel it is important to include in the manuscript. We felt that this last section was particularly important because not only did it show strong differences between the models, but it also highlighted a problem that can be overlooked in the analysis of models. There will be some elements of dynamics involved - but these differences shown (given they are radar fields plotted) are more likely to be associated with the microphysics, which certainly combined with the previous two figures and table 2 suggest a worrying factor about condensate identification and the knock on effect on the dynamics and diabatic heating in the models (and reanalyses).*

*We appreciate it is a difficult figure to interpret but we believe with the inclusion of the previous discussion on the ice water content aids the interpretation (as it essentially supports those results). We also suggest that the shape is the most important factor to consider with these plots as (as stated in the manuscript) the mask is not applied to the data given that the CFAD is a direct output from the model output field rather than crafted based on the reflectivities (as in the observations). Whilst it does show some relationship to the dynamics we feel the stronger difference (and more reason for concern) is the link with the microphysics. However, based on the links between dynamics and microphysics it does raise questions about these processes and their representation in models and re-analysis, and we would strongly encourage this type of analysis to be done for both to help improve the models and re-analysis.*

---

## Author Response (AR2)

**Response to Reviewers for WCD-2020-43**

*Dear Editor,*

*We thank both reviewers for taking the time to review and comment upon our revised manuscript. Following the comments we have made some revisions to the manuscript.*

*We provide detailed responses below with our comments and changes in green italics.*

*Kind regards,*

*David Flack, Gwendal Riviere, Ionela Musat, Romain Roehrig, Sandrine Bony, Julien Delanoe, Quitterie Cazenave and Jacques Pelon*

**Referees' comments are in black and** authors' answers in green

**Response to Anonymous Referee #1**

Thank you for addressing my concerns related to the presentation of the manuscript. The message of the paper is now clearer and the length has been reduced. Therefore, the reading of the paper was more enjoyable this time.

I recommend to publish this paper after one minor comment and a few small technical corrections have been performed.

Minor comment:

In Summary, line 424 you state that "Increasing the resolution does not increase the relative contribution of diabatic heating...", which I agree, based on Fig. 7. But based on Fig. 9 (i,j,k,l), I think that $\omega\_Q$ increases with resolution. Still, in lines 293 and 425 you state that your results disagree with Willison et al. (2013) and Trzeciak et al. (2016) papers, but I find the fact that the $\omega\_Q$ increases with resolution being quite consistent with Willison et al (2013) and Trzeciak et al. (2016) papers.

Can you please double-check the papers by Willison and Trzeciak, if they meant the increase of diabatic processes with the resolution, or the increase of the relative contribution of diabatic processes with the resolution. I think these two aspects are slightly different things, because also the dynamical parts (adiabatic processes) can increase with the resolution (and in fact does so in your results) and partly compensate for the effects of diabatic processes.

*We agree that an increase in relative contribution is not the same as just an increase. Therefore, as suggested we have double-checked the Willison and Trzeciak papers as requested. After this checking we stand by our comments that the two papers emphasize an increase of the relative contribution of diabatic processes.*

*Willison et al (2013) computed eddy heat fluxes for low and high resolution runs and found that the contribution of the eddy heat fluxes coming from diabatically-induced winds increase with resolution: p 2246 "The area average of 950–250-hPa integrated geostrophic eddy heat flux over the maritime storm track increases 24.4% when simulated at 20-km grid spacing. The area-average heat flux by the diabatic winds (y 0 dia T 0 ) increases even more dramatically at 52.6%. A relatively larger increase in diabatic flux suggests that the enhancement of diabatic effects with increased-resolution results from a strong positive feedback".*

*Trzeciak et al (2016) also looked at the relative contribution of dynamical and diabatic terms in another equation, in the pressure tendency equation. Their conclusions emphasized the weight of the diabatic term relative to the dynamical one increases with resolution. They even found that some dynamical terms are larger in coarser resolution: p 3444: "The fact that in this case the lack of diabatic contributions is compensated by larger Dφ values is remarkable and point to a stronger upper-level control of the cyclogenesis, which can be sufficiently resolved by the coarse-resolution model. ... It almost appears as if the situation*

*was so prone to the development of an intense storm that different physical pathways exist that can lead to similar cyclonic developments and that the choice of pathways is resolution-dependent." In other words, they show that in coarser resolution runs, an increase of dynamical terms may happen that compensates the decrease of diabatic terms".*

*In our case, we found an increase of the peak values of both the dynamical and diabatic terms when the resolution is increased as seen in Fig. 9. However, when spatial averages are made such an increase is less visible because the higher values cover a smaller area in high-resolution runs and this may explain why in Fig.7 we do not see a net increase of the averaged fields from low to high-resolution runs.*

*To conclude, after checking the content of the two papers, we did not change the text of our paper.*

Technical corrections:

P. 8, line 180 and 182. You use the word extratropical without dash and then with dash. Please select either of them and be consistent throughout the whole manuscript.

*Thank you for spotting this, we now use the term extratropical (without dash) throughout the manuscript.*

P. 9, line 207: "The difference occurs 18h into the hindcasts". I suggest change it to "The difference occurs 18 h after the start of the initialization" or something similar.

*This has been changed to ''The difference occurs 18 h after the start of initialisation'' as suggested.*

P. 11, line 243 and 251. Add space between "Figs." and the number.

*Thank you for spotting this, spaces have now been added in the relevant locations.*

P. 13, line 257 I'd remove the word "and", and add comma.

*Thank you, this change has now been done.*

**Response to Anonymous Referee #2**

Suggestion: minor revisions.

Lines 4,5: Sorry I missed this in the first round: what is the "c."? Why not just write it out to avoid any confusion?

*The "c." refers to circa. However, to avoid further confusion we have switched to approximately.*

Line 11: You might consider extending the other reviewer's acronym suggestion to the abstract and simply use ARPEGE and LMDZ at lines 11, 14, 15, and 16. It will be clear what you are referring to even without a "hereafter" at line 3. But, I leave it up to you.

*This decision has been made to be explicitly clear about the models (for many people ARPEGE on it's own means something different to it being defined as the climate model configuration). Therefore, in the abstract we decided to keep the full length versions within the abstract.*

Line 66, You write:
Q3 Are there any differences in the diabatic processes related to microphysical properties between the different models?

This sentence might need to be cleaned up. It could mean:

Are there any differences in the diabatic processes related to differences in microphysical properties between the two models?

Or it could mean:
Are there any differences between the two models' diabatic processes that are related to microphysical properties?

Or maybe it is something else?

*Thank you for pointing out this potential area of mis-interpretation. Your second phrasing of the question captures the intended meaning of the question so we have rephrased Q3 as such in the manuscript.*

Figure 13: This schematic is ok. The upper-level dynamical forcing shapes leave me a bit puzzled, especially at step 4. But it's your schematic, so you can decide how you'd like to keep it.

*The shapes of the upper-level forcing were to represent the shape of the PV filament in steps 2 and 3, and in step 4 changed to represent the approximate shape of the geopotential height contours representing the successive troughs. The arrow at this stage represents the direction the troughs are moving in. To aid in interpretation (of step 4) we have clarified the figure caption. The figure caption now reads:*

*"A schematic of the Stalactite Cyclone. 0) the Mesoscale Convective System that initiates the Diabatic Rossby Vortex (1) that travels along the blue arrow. The northern precursor (2) with upper-level PV cut-off that moves towards the diabatic Rossby vortex and initiates a roll-up between the two precursors at cyclogenesis to create the Stalactite Cyclone (3). Explosive deepening occurs as a result of strong diabatic heating throughout the column and the interaction with a series of embedded upper level high PV regions (the upper-level forcing here is depicted in the form of successive troughs in geopotential height moving in the direction of the white arrow; 4). Flight observations (5) indicate that ice water content is underestimated and so could have impacts on the diabatic heating and evolution of the cyclone."*